# Large volume air sample system for measuring $^{34}S/^{32}S$ isotope ratio of carbonyl sulfide

Kazuki Kamezaki[1], Shohei Hattori[1,*], Enno Bahlmann[2], Naohiro Yoshida[1,3]

[1] Department of Chemical Science and Engineering, School of Materials and Chemical Technology, Tokyo Institute of Technology, G1-17, 4259 Nagatsuta-cho, Midori-ku, Yokohama 226-8502, Japan

[2] Leibniz Centre for Tropical Marine Research, Fahrenheitstraße 6, 28359 Bremen, Germany

[3] Earth-Life Science Institute, Tokyo Institute of Technology, 2-12-1-IE-1 Ookayama, Meguro-ku, Tokyo 152-8550, Japan

*Correspondence to:* Shohei Hattori (hattori.s.ab@m.titech.ac.jp)

**Abstract**

Knowledge related to sulfur isotope ratios of carbonyl sulfide (OCS or COS), the most abundant atmospheric sulfur species, remains scarce. Earlier method developed for sulfur isotopic analysis for OCS using $S^+$ fragmentation by isotope ratio mass spectrometer is inapplicable for ambient air samples because of the large samples required (approx. 500 L of 500 pmol mol$^{-1}$ OCS). To overcome this difficulty, herein we present a new sampling system for collecting approx. 10 nmol of OCS from ambient air coupled with a purification system. Salient system features are (i) accommodation of samples up to 500 L (= approx. 10 nmol) of air at 5 L min$^{-1}$, (ii) portability of adsorption tubes (1/4 inch (0.64 cm) outer diameter, 17.5 cm length, approx. 1.4 cm$^3$ volume) for preserving the OCS amount and $\delta^{34}S(OCS)$ values at −80 °C, respectively, for up to 90 days and 14 days, and (iii) purification OCS from other compounds such as $CO_2$. We tested the OCS collection efficiency of the systems and sulfur isotopic fractionation during sampling. Results show precision ($1\sigma$) of $\delta^{34}S(OCS)$ value as 0.4 ‰ for overall procedures during measurements for atmospheric samples. Additionally, this report presents diurnal variation of $\delta^{34}S(OCS)$ values collected from ambient air at Suzukakedai campus of Tokyo Institute of Technology located in Yokohama, Japan. The observed OCS concentrations and $\delta^{34}S(OCS)$ values were, respectively, 447–520 pmol mol$^{-1}$ and from 10.4 ‰ to 10.7 ‰ with a lack of diurnal variation. The observed $\delta^{34}S(OCS)$ values in ambient air differed greatly from previously reported values of $\delta^{34}S(OCS) = (4.9 \pm 0.3)$ ‰ for compressed air collected at Kawasaki, Japan, presumably because of degradation of OCS in cylinders and collection processes for that sample. The difference of atmospheric $\delta^{34}S(OCS)$ values between 10.5 ‰ in Japan (this study) and ~13 ‰ recently reported in Israel or Canary Islands, spatial and temporal variation of $\delta^{34}S(OCS)$ values are expected to be a link between anthropogenic activities and OCS cycles. The system presented herein is useful for application of $\delta^{34}S(OCS)$ for investigation of OCS sources and sinks in the troposphere to elucidate its cycle.

# 1    Introduction

Carbonyl sulfide (OCS) is the most abundant sulfur-containing gas in ambient air with atmospheric concentrations of approx. 500 pmol mol$^{-1}$ in the troposphere (Chin and Davis, 1995; Montzka et al., 2007). In fact, OCS can be transported to the stratosphere because the average residence time of OCS is longer than two years (Brühl et al., 2012). In the stratosphere, it

is converted to stratospheric sulfate aerosols (SSA) through atmospheric sink reactions (Crutzen, 1976). Therefore, OCS must be regarded as an important sulfur source for SSA, playing an important role in the Earth's radiation budget and in ozone depletion. Moreover, because leaves consume OCS whenever assimilating $CO_2$, but do not emit OCS to the atmosphere by respiration (Sandvalo-Soto et al., 2005), OCS can be a tracer of gross primary production (GPP) on land (Campbell et al., 2008). For those reasons, elucidating the OCS dynamics in the atmosphere is important to elucidate the

carbon cycle. Nevertheless, tropospheric OCS sources and sinks entail great uncertainty (Watts, 2000; Kremser et al., 2016) because of missing sources in the atmospheric budget of 230–800 Gg a$^{-1}$ S equivalents as revealed by top-down modelling (Berry et al., 2013; Glatthor et al., 2015; Kuai et al., 2015).

Isotope analysis is a useful tool to trace sources and transformations of trace gases (Johnson et al., 2002; Brenninkmeijer, 2003). To quantify OCS sources and sinks in natural environments using isotope analysis, determination of

isotopic fractionation for reactions and ambient measurements is required. To date, isotopic fractionations occurring in the reactions of OCS have been determined for almost all OCS sink reactions in the stratosphere: OCS photolysis (Hattori et al., 2011; Lin et al., 2011; Schmidt et al., 2013) as well as reactions with OH (Schmidt et al., 2012) and O($^3$P) (Hattori et al., 2012). Furthermore, the sulfur isotopic fractionation during soil bacterial degradation and enzymatic degradation were ascertained based on laboratory experiments (Kamezaki et al., 2016; Ogawa et al., 2017). Based on the analysis of

commercially available compressed air our group suggested a $\delta^{34}S$ value of (4.9 ± 0.3) ‰ for tropospheric OCS (Hattori et al., 2015). However, very recently, Angert et al. (2019) reported a markedly different $\delta^{34}S$ value of (13.1 ± 0.7) ‰ for tropospheric OCS using a gas chromatograph (GC)/multi collector-inductively coupled plasma mass spectrometer (MC-ICP-MS). For the measurement of sulfur isotope ratios ($\delta^{33}S$, $\delta^{34}S$ and $\Delta^{33}S$ values) of OCS in our laboratory, an online method measuring on a GC/isotope ratio (IR)-MS using S$^+$ fragmentation ions had been developed (Hattori et al., 2015). This

method supports simple analysis of sulfur isotopic compositions of OCS over 8 nmol. However, application this method for atmospheric samples has yet to be done by this GC/IR-MS method, because of the large sample amounts that are necessary (i.e. 500 L of 500 pmol mol$^{-1}$ OCS). Therefore, we aimed to develop a large volume air sampling system to apply S$^+$ IR-MS method for atmospheric samples.

To date, several methods have been developed for concentration measurements using grab samples of air coupled

with sampling/purification systems in the laboratory (e.g. Inomata et al., 1999; Xu et al., 2001; Montzka et al., 2004; Kato et al., 2012). Most systems collect 2–5 L of atmospheric samples for measuring OCS concentrations. The collected OCS is extracted in the laboratory with a combination of adsorbents at subambient temperatures: Tenax TA with dry ice/methanol (Inomata et al., 1999) or dry ice/ethanol (Hattori et al., 2015), glass beads with liquid oxygen (Montzka et al., 2004) or liquid

argon (Xu et al., 2001), 2,3-Tris (2-cyanoethoxy) propane with liquid oxygen (Kato et al., 2012). Grab sampling, however, is unrealistic when collecting 500 L of air. Therefore, we developed a large-volume air sampling system for measuring sulfur isotope ratios of OCS. We modified a large-volume air sampling system developed for carbon isotope measurement for halocarbons such as chloromethane and bromomethane, which have concentrations of pmol mol$^{-1}$ level in ambient air

(Bahlmann et al., 2011). Subsequently, we combined this sampling system and newly developed an online OCS purification system for separation from impurities such as $CO_2$, which is $10^6$ times more abundant in air than OCS. For the current study, we describe the systems and its applications to atmospheric observation. We provide first results for diurnal variations of $\delta^{34}S(OCS)$ in ambient air from samples collected at Suzukakedai campus of the Tokyo Institute of Technology located in Yokohama, Japan.

## 2        Materials and methods

### 2.1 Samples

An overview of the synthetic samples used for method evaluation in this study is given in Table 1. Commercial samples containing 10.5 % OCS balanced with high-purity He as sample A (99.99995 % purity; Japan Fine Products Co. Ltd., 

Kawasaki, Japan) and 5.9 µmol mol$^{-1}$ OCS balanced with high-purity He as sample B (99.99995 % purity; Japan Fine Products Co. Ltd.) were used (Table 1). Furthermore, we synthesized OCS from three kinds of sulfur powders, designated as sample C produced from sulfur power (99.99 % purity; Fujifilm Wako Pure Chemical Corp., Japan), sample D produced from sulfur powder (99.98% purity; Sigma-Aldrich Corp. LLC, Missouri, USA), and sample E (a mixture of sulfur powders used for samples C and D) with a reaction with CO (99.99 % purity; Japan Fine Products Co. Ltd., Kawasaki, Japan) in a 

similar manner to that described by Ferm (1954) and Hattori et al. (2015) (Table 1). The OCS concentrations for samples A and B were determined against to the in-house synthesized OCS (i.e. 100 %) diluted to 10 % using high-purity He (99.99995 % purity; Japan Fine Products Co. Ltd.). It is noteworthy that the OCS concentration in sample B had showed no change at least four years after the publication of Hattori et al. (2015).

For the testing of repeatability and collection efficiency of the systems, we used three commercially available 

cylinders of compressed air samples in a collected in Kawasaki, Japan (Toho Sanso Kogyo Co., Ltd., Yokohama, Japan), sample F (collected on 25 July 2017), sample G (collected on 2 July 2012), sample H (collected on 2 December 2017), sample I (collected on 26 October 2018), sample J (collected on 1 December 2018), and sample K (collected on 26 December 2018) (Table 2). These compressed air samples in these cylinders are collected by compressor (YS85-V; Toa Diving Apparatus Co., Ltd., Tokyo, Japan) and are not dried. Sample G was used as sample E for Hattori et al. (2015). Its 

$\delta^{34}S(OCS)$ value was (4.9 ± 0.3) ‰. It was postulated as the global representative value at that moment. All compressed air cylinders are made of manganese steel without special wall treatments, engendering concerns about OCS decomposition in the cylinders.

## 2.2 Sampling system

A schematic diagram of the sampling system is depicted in Figure 1. The sampling system size and weight are 50 cm × 50 cm × 50 cm (width × height × depth), and 4 kg except for a dewar (37 cm outer diameter, 66 cm height and 11 kg weight) (MVE SC 20/20; Chart Industries Inc., Georgia, USA). For field campaigns, the system can be easily disassembled and transported in two containers of 40 cm × 30 cm × 20 cm (width × height × depth). Reassembling the sampling system on site can easily be done within 2 h making it suitable for field campaigns. Main compartments of the sampling system are 1/4 inch (0.64 cm) PTFE tubes, 1/8 inch (0.32 cm) stainless steel tubes, 1/16 inch (0.16 cm) Sulfinert-treated stainless steel tubes (Restek Corp., PA, USA), Sulfinert-treated stainless steel ball valves V1, V2, V5, and V9, and stainless steel ball valves V3, V4, V6, V7, and V8 behind the sampling tube (Figure 1). Excluding union tees made of stainless steel immediately before the pump, union tees coming in contact with the sampled OCS are made of Sulfinert-treated stainless steel (Figure 1).

The cryotrap sampling tube for OCS concentration from ambient air consists of an outer stainless steel tube (3/4 inch (1.9 cm) outer diameter, 50 cm length) with an air inlet at the side 4 cm below the top and an inner 1/4 inch (0.63 cm) stainless steel tube (Bahlmann et al., 2011). From top to bottom, the sampling tube package is the following: 0–30 cm, empty; 30–40 cm, silanized glass beads 2 mm; 40–43 cm, Tenax TA (60/80 mesh; GL Science Inc., Tokyo, Japan); 43–47 cm, Porapak N (80/100 mesh; Sigma-Aldrich Corp., Japan); 47–50 cm, empty, and adsorbents separated by plugs of precleaned glass wools (GL Science Inc., Tokyo, Japan). We developed this sampling tube according to Bahlmann et al. (2011). Detailed functions of respective components are described therein. Briefly, the glass bead traps the remaining water vapor from the sampled air and prevents water vapor adsorption on the Tenax TA and Porapak N. The glass bead further increases the temperature exchange between the cryotrap walls and the sampled air. The Tenax TA and Porapak N can be used for trapping volatile organic compounds. We assume that OCS is sampled on the Tenax TA and Porapak N, but most of OCS might be trapped on Tenax TA. Although some components might not be necessary for OCS collections, up to this point, it is working well for OCS sampling.

The adsorption tube consists of a stainless steel tube (with 1/4 inch (0.63 cm) outer diameter, 17.5 cm length) filled with Tenax TA. Before experiments, the sampling tube and the adsorption tube were conditioned in the laboratory, respectively, using 100 mL min$^{-1}$ high-purity He flow at 160 °C with an electric heating mantle (P-22; Tokyo Technological Labo Co., Ltd., Kanagawa, Japan) for 6 h and 50 mL min$^{-1}$ high-purity He flow at 330 °C by an electric heating mantle (P-25; Tokyo Technological Labo Co., Ltd.) for 6 h. We confirmed that possible contamination of OCS in the tubes was less than 10 pmol after conditioning. We also confirmed that the surface was inert at least three days and that the inactive state of the surface of adsorbents in these tubes would be maintained under a no-leakage condition. It is noteworthy that conditioning steps would be required if stainless tubes are replaced by Sulfinert-treated tubes/valves because this conditioning was aimed at removing strongly adsorbed volatile organic compounds such as ethanol and acetaldehyde in adsorbents, which might interfere with OCS collection and/or react with OCS.

During sampling, the valves V1, V2, V3 and V4 were opened. Then atmospheric samples were drawn with a low volume diaphragm pump (LV-40BW; Sibata Scientific Technology Ltd., Saitama, Japan) through the sampling system with flow of $(5.00 \pm 0.25)$ L min$^{-1}$. The air was first passed through a membrane filter (47 mm diameter, 1.2 μm pore, Pall Ultipor N66 sterilizing-grade filter; Pall Corp., New York, USA) set in a NILU filter holder system (70 mm diameter, 90 mm length: Tokyo Dylec Corp., Tokyo, Japan) to remove atmospheric aerosol. Then it was directed through a condenser (EFG5-10; IAC Co. Ltd., Japan) kept at approximately −15 °C to remove water vapor from the air. The air was then passed through the sampling tube at temperatures of −140 to −110 °C by vapor of the liquid $N_2$ in a dewar. The OCS was enriched in the sampling tube, whereas other main gases ($N_2$, $O_2$, Ar, etc.) were passed through the sampling tube.

After sampling, the valves V1 and V4 were closed; and the valves V5, V6, V7, and V8 were opened. Then, the sampling tube was removed carefully from the dewar manually and was heated gradually to 130 °C. The vaporized gases in the sampling tube were passed to the adsorption tube cooled at −78 °C using dry ice/ethanol after removal of the remaining water vapor by a Nafion dryer (MD-110-96S; Perma Pure LLC, NJ, USA). The flow rate was regulated (approx. 50 mL min$^{-1}$) by a needle valve equipped with a flow meter for 20 min. After the flow rate became lower than 10 mL min$^{-1}$, V4 was opened. The sampling tube was flushed with pure $N_2$ (>99.99995 vol. %) with 50 mL min$^{-1}$ for 40 min. After the transfer of samples, V6, V7 and V8 were closed. Then OCS was preserved in the adsorption tube. We initially confirmed that OCS did not pass through an adsorption tube at a flow rate lower than 50 mL min$^{-1}$ using two adsorption tubes connected in series from the second adsorption tube: OCS was observed only from the first tube; not from the second tube. For this study, the collected OCS samples in adsorption tubes were measured within 30 min, except for the preservation test.

## 2.3 Purification system

After sampling OCS from the air using the sampling system as described above, the collected OCS was purified and connected directly to the measurement system. The schematic system is shown in Figure 2. Excluding a fused silica capillary tube, all tubes and valves are made of stainless steel. U-shaped trap 1 is a 50 cm, 1/4 inch (0.64 cm) outer diameter (1/8 inch (0.32 cm) inner diameter) stainless steel tube. U-shaped trap 2 is a 30 cm, 1/8 inch (0.32 cm) outer diameter (1/16 inch (0.16 cm) inner diameter) stainless steel tube filled with Tenax TA (60/80 mesh; GL Science Inc.). Before the experiment, trap 2 is heated to 150 °C by an electric heating mantle (P-22; Tokyo Technological Labo Co., Ltd.) for 30 min at 30 mL min$^{-1}$ with high-purity He for conditioning. Coil-shaped trap 3 is an empty stainless steel tube (1/16 inch (0.16 cm) outer diameter, 50 cm length). Coil-shaped trap 4 is a fused silica capillary tube (0.32 mm inner diameter, 50 cm length, GL Science Inc.). The GC1 (GC-8610T; JEOL Ltd., Tokyo, Japan) is equipped with a column packed with Porapak Q (80/100, GL Science Inc.) (1/8 inch (0.32 cm) outer diameter, 3 m length) to separate OCS from $CO_2$. The GC1 oven temperature for OCS purification was programmed to provide 30 °C for 5 min, ramping to 60 °C at 30 °C min$^{-1}$, followed by ramping to 230 °C at 30 °C min$^{-1}$ from 40 min, and 230 °C for 1 min.

After the adsorption tube containing OCS was connected to the purification system, v3, v4, and v5 were opened and the air in the line was pumped out using a rotary pump (DA-60D; Ulvac Kiko, Miyazaki, Japan) for 5 min; v3, v4, and v5 were then closed. When the adsorption tube was heated at 130 °C and v2, v7, v8, and v6 were opened, gases in the adsorption tube passed through trap 1 cooled by dry ice (−78 °C) to remove trace remnant water vapor. Also, OCS was collected in trap 2, with Tenax TA cooled by dry ice/ethanol (−72 °C) with high-purity He flow rate of 30 mL min$^{-1}$. After 15 min, port valve (PV) 1 was changed. Trap 2 was then removed from dry ice/ethanol and was heated at 130 °C. The retention times of $CO_2$ and OCS were initially determined by injecting a mixture of 8 mmol of $CO_2$ from pure $CO_2$ in a cylinder (99.995 % purity; Japan Fine Products Co. Ltd.) and 10 nmol of OCS from sample C. They were, respectively, 3–10 min for $CO_2$ and 20–30 min for OCS at a flow rate of 25 mL min$^{-1}$. Trap 3 was cooled by liquid $N_2$ from 10 min; PV2 was changed from 15 min to 35 min after injection of samples to GC1 to introduce OCS to trap 3. OCS with high-purity He was passed through the column and collected in trap 3 for 20 min. After OCS collection in trap 3, the OCS was again transferred to trap 4 in liquid $N_2$ at 6 mL min$^{-1}$ by high-purity He with removal of liquid $N_2$ from trap 3 to a cryofocus. Trap 4 was then removed from liquid $N_2$; the OCS passed through the GC2 and introduced directly to the detectors (quadrupole mass spectrometer (Q-MS) or IR-MS depending on the experiments explained below).

**2.4 Determination of the OCS concentration**

The OCS concentrations were measured by GC/Q-MS (7890A; Agilent Technologies Inc., CA, USA coupled to Q-MS, 5975C; Agilent Technologies Inc., CA, USA) equipped with a capillary column (0.32 mm inner diameter, 25 m length, and 10 μm thickness; HP-PLOT Q, Agilent Technologies, CA, USA). The He flow was set to 1.5 mL / min and oven temperature program was set as 60 °C for 15 min, ramped to 230 °C at 60 °C min$^{-1}$, then 230 °C for 1 min.

To ascertain the OCS concentration of samples A once a month, and to ascertain the collected OCS amounts using a sampling system, we made a calibration curve for OCS ranging 0.1 nmol to 10 nmol using Q-MS. The calibration curve for the nanomole level is made by injection of sample B with a volume of 0.5 mL, 2.2 mL, 4.4 mL, 8.8 mL, 11 mL, 13.2 mL, 17.6 mL, 22 mL, and 44 mL ($n = 3$). The precision (standard deviation ($1\sigma$) relative to mean) of the OCS amount by a syringe injection was estimated ± 3 % by the standard deviation of the relative error between the measured values and the estimated value for calibration curves.

To ascertain the OCS concentrations of samples F and G, we prepared calibration curves for OCS ranging 0 pmol to 100 pmol using Q-MS. Calibration curve for the picomole level is made by injection of sample B with volume of 0 μL, 200 μL, 400 μL, and 800 μL ($n = 3$) with precision ± 3 % as estimated similarly above. For determination of OCS concentrations of samples F and G, samples F and G were stored in 50 mL two-neck glass bottles with atmospheric pressure and were introduced to the purification system from an attached glass bottle instead of an adsorption tube. The measured OCS concentrations for samples F and G were, respectively, (380 ± 15) pmol mol$^{-1}$, and (160 ± 5) pmol mol$^{-1}$ (Table 2).

The OCS concentrations for the samples F, G, H and I were lower than that of typical atmospheric OCS concentrations (400–550 pmol mol$^{-1}$) (Montzka et al., 2007), even though the samples were compressed air collected from the ambient atmosphere. Because we were concerned about the changes in OCS concentrations for the samples F and G, the OCS concentrations for the samples F and G were measured at least within a week before or after the experiment. In a similar manner, the cylinders of sample H, I, J, and K were used for experiment within 2–3 days. Therefore, the change of OCS concentration in samples might occur.

## 2.5 Determination of the sulfur isotope ratios of OCS

For determination of the sulfur isotope ratios of OCS, OCS was passed through the GC2 after a purification system as described above. Then it was introduced directly to the IR-MS (MAT253; Thermo Fisher Scientific Inc., Berlin, Germany) via an open split interface (ConFlo IV; Thermo Fisher Scientific Inc.). Reference OCS of sample A was purified with liquid N$_2$ (−196 °C) and then introduced via a conventional dual inlet system. Pure OCS is not commercially available in Japan because of its toxicity (Hattori et al. 2015). In addition to the method introducing OCS to IR-MS as described above, conventional syringe injection line, which was previously used for Hattori et al. (2015) and Kamezaki et al. (2016), was also used for comparison or calibration. Briefly, the syringe-injected OCS was collected in stainless steel tubes (10.5 mm inner diameter, 150 mm length) cooled at −196 °C by liquid N$_2$ with vacuum by a rotary pump (Pascal 2010; Pfeiffer Vacuum GmbH, Aßlar, Germany) gently with regulation using a valve. After transfer of OCS to the trap, the two-way six port valve was changed. Then liquid N$_2$ was removed from the trap. Subsequently, OCS was transferred and collected in a fused silica capillary tube (0.32 mm inner diameter, 50 cm length; GL Science Inc.) covered by a stainless steel tube containing liquid N$_2$ for 13 min before being introduced into the GC/IR-MS system.

In the IR-MS ion source, electron impact ionization of OCS produced S$^+$ fragment ions. The sulfur isotope ratios in OCS were therefore determined by measuring the fragment ions $^{32}$S$^+$, $^{33}$S$^+$, and $^{34}$S$^+$ using triple Faraday collector cups. The typical precisions ($1\sigma$) of the replicate measurements ($n$ = 3) are, respectively, 0.4 ‰, 0.2 ‰, and 0.3 ‰ for $\delta^{33}$S(OCS), $\delta^{34}$S(OCS), and $\Delta^{33}$S(OCS) values. A reference OCS gas was introduced for 20 s at three times started at $t$ = 350 s, 825 s, and 1025 s. The reference gas at $t$ = 350 s was used as the reference for all calculations of OCS sulfur isotope ratios. To remove hydrogen sulfide and ethane from ambient samples, from $t$ = 300 s, the effluent from the GC column was kept off the MS line using back-flushed helium flow. Sulfur isotope ratios are typically reported as

$$\delta^x S = {}^x R_{sample} / {}^x R_{standard} - 1, \qquad (1)$$

where $^x R$ represents the isotopic ratios ($^x$S/$^{32}$S, where $x$ = 33 or 34) of the samples and standards. The sulfur isotope ratios are reported relative to the Vienna Canyon Diablo Troilite (VCDT, quoted as per mil values (‰). In addition to the $\delta$ values, capital delta notation ($\Delta^{33}$S value) is used to distinguish mass-independent fractionation (MIF; or non-mass-dependent

fractionation) of sulfur, which causes deviation from the mass-dependent fractionation (MDF) line. The $\Delta^{33}S$ value describes the excess or deficiency of $^{33}S$ relative to a reference MDF line. It is expressed as

$$\Delta^{33}S = \delta^{33}S - [(\delta^{34}S + 1)^{0.515} - 1]. \qquad (2)$$

The $\delta$ values in this study were determined using the following processes. First, we ascertained the $\delta^{34}S$ value of sample A by converting OCS to $SF_6$. The $\delta^{34}S(SF_6)$ value was measured relative to the VCDT scale by comparing $SF_6$ similarly converted from IAEA-S-1 ($Ag_2S$: $\delta^{34}S = -0.30$ ‰; Robinson, 1993) as described by Hattori et al. (2015). The measured $\delta^{34}S$ value of sample A was 12.6 ‰, which was lower than the data presented by Hattori et al. (2015) with 14.3 ‰ (Table 1). Secondly, the $\delta^{34}S$ value of sample B, which was used as working standard for $\delta^{34}S$ measurements, were ascertained from comparison with the $\delta^{34}S$ value (in VCDT scale) of sample A in GC/IR-MS method using $S^+$ fragment ion. The $\delta^{34}S(OCS)$ value of sample B in this study was $(14.1 \pm 0.2)$ ‰ (Table 1), showing no significant difference with the $\delta^{34}S(OCS)$ value of sample B $(14.3 \pm 0.2)$ ‰ in data presented by Hattori et al. (2015). It is noteworthy that we also found that the OCS concentration in sample B was not changed. Sample B was used as the daily working standard for GC/IR-MS measurement to ascertain sample $\delta^{34}S(OCS)$ values for other samples used throughout this study (Table 1).

## 3        Results and Discussion

### 3.1 Sampling efficiency of OCS

To test the sampling and desorption efficiency, the cylinder containing sample F was connected to a flow meter and the flow was adjusted to 6 L min$^{-1}$ with a needle valve. 5 L min$^{-1}$ were drawn through the sampling system with a pump and the remainder was vented into the air to maintain atmospheric pressure at the sampling inlet. The samples were collected within two days to prevent OCS loss in the cylinder. The vent flow was measured with a flow meter (ACM-1A; Kofloc, Tokyo, Japan). To ascertain the trapping efficiency OCS was sampled for 10 min, 50 min, and 100 min with blank test intervals as presented in Figure 3a (see section 2.2 for sampling procedure). The sampling times corresponded to sampling volumes of $(50 \pm 2.5)$ L, $(250 \pm 13)$ L and $(500 \pm 25)$ L and the respectively and the corresponding OCS amounts were $(0.77 \pm 0.04)$ nmol, $(3.9 \pm 0.2)$ nmol, and $(7.7 \pm 0.4)$ nmol respectively (Figure 3a).

Recovery and precision ($1\sigma$) for OCS amounts collected for sampling times of 10 min, 50 min, and 100 min were, $(0.9 \pm 0.1)$ nmol ($n = 3$), $(3.6 \pm 0.2)$ nmol ($n = 3$) and $(7.4 \pm 0.3)$ nmol ($n = 2$), respectively. The OCS blanks were smaller than 15 pmol. These results indicate that yield of OCS during sampling and transferring from the sampling tube to the adsorption tube is almost over 95 %. The memory effect of the system between the sampling run is expected to be less than 1% when sampling OCS amount over 3 nmol (approx. 50 min). Figure 3b presents a comparison of OCS amount between observed OCS amounts and OCS amounts calculated based on OCS concentration in sample F and sampling time, showing that all results fall on the 1:1 line. This suggests that almost 100 % of OCS for sampling runs were collected in the sampling

tube and were transferred successfully to the adsorption tube. Although the collected OCS amount in 10 min was slightly larger than the expected OCS amount, the OCS amounts in 100 min were slightly lower than the expected OCS amount. This result indicates that a small OCS contamination during the sampling and a purification system might exist but that it might not be significant, as discussed above.

## 3.2 Accuracy of the sulfur isotopic analysis of OCS via sampling/purification systems

In the developed system, the possibility exists that OCS is lost by passing OCS through GC1. Also, because the flow rate of approx. 50 mL / min was lower than the flow rate of approx. 200 mL / min reported by Hattori et al. (2015), the possibility exists that OCS was lost by Trap 1. Therefore, to assess these possibilities, the following test was conducted. Firstly, 5 nmol of OCS was injected to a system consisting of Trap 2, GC2, and Trap 4 and measured as true value. Then, the same amount of OCS was introduced into the developed purification system and the amount of OCS obtained was compared to true value. These tests revealed an OCS loss of less than 2 % using a newly developed method and suggest a complete recovery of OCS within the given limits of uncertainty ($\pm$ 3 %). To assess the dependence of the sulfur isotopic measurements on the OCS amount, different amounts of OCS using sample B were tested. We introduced aliquots of 3 nmol, 6 nmol, 10 nmol, and 15 nmol of sample B over 30 min with a gastight syringe via a syringe port made from a tee union with a septum. The syringe port was place between the inlet filter and the condenser and the sampling inlet was connected to of high-purity $N_2$ ($>$ 99.99995 vol. %; Nissan Tanaka Corp., Saitama, Japan) (Figure 1). For each experiment, a total volume of 500 L $N_2$ was processed. The OCS contamination for this experiment was $(0.30 \pm 0.16)$ nmol ($n = 3$) when we flushed with 500 L of pure $N_2$ stream. For comparison, similar amounts of OCS were also injected using a syringe injection system developed previously (Hattori et al., 2015). Comparisons of OCS concentrations and $\delta$ and $\Delta$ values are depicted in Figure 4. Although the observed OCS isotope ratios using 3 nmol OCS with the developed method were scattered ($1\sigma$ uncertainty: 1.0 ‰, 1.0 ‰, and 0.5 ‰, respectively, for $\delta^{33}S(OCS)$, $\delta^{34}S(OCS)$, and $\Delta^{33}S(OCS)$ values), the reproducibilities at the 6 nmol level were sufficient ($1\sigma$ uncertainty: 0.4 ‰, 0.2 ‰, and 0.4 ‰, respectively, for $\delta^{33}S(OCS)$, $\delta^{34}S(OCS)$, and $\Delta^{33}S(OCS)$ values) and were similar to those obtained with the conventional syringe injection system for Hattori et al. (2015) (Figure 4). Consequently, this system better accommodates OCS samples over 6 nmol, indicating the necessity for collection of ambient air in amounts greater than 300 L.

Furthermore, to test possible sulfur isotopic fractionations during sampling/purification processes, which might change the measurement accuracy, we compared the developed sampling/purification system with the conventional syringe injection system using 8 nmol of the in-house synthesized OCS (samples B, C, D, and E) with triplicates injections. In Figure 5, the $\delta^{34}S(OCS)$ values measured using the developed sampling/purification system were 0.2 ‰ lower (sample B) but 0.8 ‰, 0.4 ‰, and 0.6 ‰ higher (samples C, D, and E, respectively) than those measured using the syringe injection system of Hattori et al. (2015) (Table 1). This phenomenon was observed similarly for the $\delta^{33}S(OCS)$ values (Figure 5c), indicating that this process is not isotopic fractionation but that rather suggests contamination during the sampling processes. When

considering $(0.30 \pm 0.16)$ nmol OCS (i.e. approx. 4 % for 8 nmol OCS samples) with $\delta^{34}$S of 3–18 ‰ covering reported $\delta^{34}$S range of OCS sources (Newman et al., 1990), the accuracy of the $\delta^{34}$S(OCS) can be shifted −0.3 to +0.3 ‰. Because the precision of 1$\sigma$ uncertainty is 0.2 ‰, the overall precision values (1$\sigma$) for $\delta^{34}$S of this sampling/purification system were estimated as 0.4 ‰.

### 3.3 Sulfur isotope ratio for atmospheric OCS

Four ambient air samples were collected at the Suzukakedai campus of Tokyo Institute of Technology located in Yokohama, Japan (35.5°N, 139.5°W, 27 m height) during 23–25 April 2018 every 12 h (sampling times were 23 April 2018 12:00, 24 April 2018 00:00, 24 April 2018 12:00, and 25 April 2018 00:00). The sampling volume was 500 L (i.e. 100 min with a pump flow of 5 L min$^{-1}$). Measurements of OCS concentrations and sulfur isotope ratios were carried out within 30 min after the sampling. The time for single measurement of $\delta^{34}$S value for atmospheric OCS was 100 min (500 L) for sampling of air, 40 min for transferring to the absorption tube, 40 min for purification, and 20 min for measurement using IR-MS. The OCS concentrations and $\delta^{34}$S(OCS) values observed for ambient air are presented in Figure 6.

In contrast to the $\delta^{34}$S(OCS) value, the $\delta^{33}$S(OCS) value in air was not determined because of the unexpected peak (approx. 40 mV height) observed for $m/z$ 33, which slightly overlapped the OCS peak of the chromatogram (Figure 7). We notably didn't observe any interferences on $m/z$ 32 and $m/z$ 34. The interfering compound could have not yet been identified. Known fragments interfering on $m/z$ 33 are $CH_5O^+$ originating from the protonation of methanol and/or the reaction of $CH_3^+$ with $H_2O$, $CH_2F^+$ that is indicative hydrofluorocarbons, and/or $NF^+$ deriving from nitrogen trifluoride ($NF_3$). To measure $m/z$ 33 of OCS without interferences, further improvement of peak separation of OCS with interferences is required by changing parameter of the separation in the system and/or data processing. For example, custom made MATLAB routine, which can extrapolate the peak tail of interference via an exponentially decaying function to distinguish the two gaseous species as described in Zuiderweg et al. (2013), would enable us to analyze $m/z$ 33 in addition to the standard ISODAT software used for isotope ratio measurements.

The observed OCS concentrations for atmospheric samples were 447–520 pmol mol$^{-1}$ (Figure 6a), averaging $(492 \pm 34)$ pmol mol$^{-1}$. Data show no clear differences between 0:00 and 12:00 in two days ($p$-value = 0.65). This OCS concentration observed at Suzukakedai campus shows good agreement with the OCS concentrations observed at similar latitude in USA (e.g. 400–550 pmol mol$^{-1}$; Montzka et al., 2007). Berkelhammer et al. (2014) reported diurnal variation for OCS concentrations in USA with the lowest at 8:00 and the highest at 16:00 with 80 pmol mol$^{-1}$ changes in a day. Moreover, the differences of OCS concentrations for four atmospheric samples were less than 80 pmol mol$^{-1}$. The observed $\delta^{34}$S(OCS) values of four atmospheric samples were 10.4–10.7 ‰ (Figure 6b) and averaged $(10.5 \pm 0.4)$ ‰. The $\delta^{34}$S(OCS) values also showed no clear diurnal difference ($p$-values = 0.29) (Figure 6b). Given the diurnal OCS variations, some future study is clearly necessary to ascertain whether or not $\delta^{34}$S(OCS) values have diurnal variations by comparing $\delta^{34}$S(OCS) values for the highest OCS concentration at 8:00 and the lowest OCS concentration at 16:00.

It is noteworthy that the $\delta^{34}$S(OCS) values of four atmospheric samples were clearly distinct from our earlier observed $\delta^{34}$S(OCS) value of (4.9 ± 0.3) ‰ obtained from sample G (Hattori et al., 2015), that was postulated as a global representative $\delta^{34}$S(OCS) value in the atmosphere. In fact, the OCS concentrations in the commercial cylinders F, G and H were significantly lower than typical atmospheric OCS concentrations of approx. 500 nmol mol$^{-1}$ (Table 2). Ascertaining the

$\delta^{34}$S(OCS) value in sample G using the current sampling/purification system yielded a $\delta^{34}$S(OCS) value of (6.1 ± 0.2) ‰ being slightly higher than the previous value of (4.9 ± 0.3) ‰ (Hattori et al., 2015). It is possible to explain this 1.2 ‰ increase for $\delta^{34}$S(OCS) value for a case in which the contaminated OCS has $\delta^{34}$S(OCS) value with over 17 ‰. However, such a high $\delta^{34}$S(OCS) value from contamination requires a situation in which the contaminated OCS come only from the ocean, which is not likely. Because the atmospheric $\delta^{34}$S(OCS) values in this study were (10.5 ± 0.4) ‰ and higher than that

for sample G, the increased $\delta^{34}$S(OCS) values are expected to be affected by isotopic fractionation during OCS degradation in the cylinder and not by contamination. The causes for the OCS losses in the commercial pressurized air cylinders could not be investigated here. Indeed, as reported by Kamezaki et al. (2016), OCS is decomposed by hydrolysis, which increases the $\delta^{34}$S(OCS) value. Additionally, observation of OCS loss caused by adsorption to walls in the canister was reported by Khan et al. (2012). The compressed air of samples F and G might be affected by anthropogenic OCS sources at the sampling

site and/or during the compressing processes. All in all, the $\delta^{34}$S(OCS) value of sample G is no longer considered as a representative of atmospheric OCS.

**3.4 Preservation tests**

As described above, we measured OCS concentration and sulfur isotope ratio of atmospheric samples within 30 min after sampling. The OCS concentrations are consistent with the observed OCS concentrations in the same latitude and our

tests revealed no OCS losses under these conditions. However, after the development of the system, we realized up to 50 % of OCS can be decomposed during storage of the adsorption tube after we have measured the samples within 14 days after sampling (Figure 8a). We also found that the OCS in the stainless steel adsorption tubes stored at 25 °C showed only slight changes in concentration with (−6 ± 6) % and for $\delta^{34}$S(OCS) values with (0.2 ± 0.4) ‰ after 3 h (Figure 8b). All data sets presented up to this point were undertaken immediately after the sampling (i.e. shorter than 30 min.). Therefore, we did not

expect marked changes in OCS concentrations and the $\delta^{34}$S(OCS) values for most datasets including atmospheric OCS samples. Because OCS is known to react with the surface of stainless steel (Khan et al., 2012), for future use, this fact requires appropriate ways of preservation of OCS during transportation from field sampling sites to laboratory until analysis.

In order to minimize potential OCS decomposition on the surface wall, we modified the adsorption tube by replacing the stainless steel tube and valves by a Sulfinert-treated tube and Sulfinert-treated valves. The preservation of OCS

on the modified sampling tubes at different storage temperatures using samples H, I, J and K. The samples were processed as that described in section 2.2 and transferred to the adsorption tubes. The adsorption tubes were stored at temperatures of

25 °C, 4 °C, −20 °C and −80 °C, respectively, until measurements. After each storage period, the samples were analyzed for OCS yields and $\delta^{34}$S(OCS) values as described in sections 2.3, 2.4, and 2.5. A rapid OCS decomposition of approximately 20 % during 7 days of storage was observed for the stainless steel adsorption tubes stored at 25 °C. A similar pronounced loss was observed for the Sulfinert-treated adsorption tubes stored at 4 °C but at a storage temperature of −20 °C. The OCS

was stable for 30 days at −20 °C, and for at least 90 days at −80 °C within 1$\sigma$ uncertainty of 6 % (Figure 8a). Furthermore, we found that the $\delta^{34}$S(OCS) values showed no significant change during storage for at least 14 days at −80 °C (Figure 8b). These results demonstrate that it is possible to apply this method for field campaigns by storing the adsorption tube at −80 °C after sampling.

## 3.5 Atmospheric implications

The $\delta^{34}$S(OCS) value of (10.5 ± 0.4) ‰ is generally consistent with earlier estimation by Newman et al. (1991), who expected the mean $\delta^{34}$S(OCS) values of 11 ‰ based on the flux of continental emission as 3 ‰ and oceanic emission as 18 ‰ (Newman et al., 1991). This estimation is based on older information, but current measurements of atmospheric DMS and DMSP are similar to 18 ‰ (Said-Ahmad and Amrani, 2013; Amrani et al., 2013; Oduro et al., 2012); continental sulfur sources also show approx. 0–5 ‰ (Tcherkez and Tea, 2013).

It is noteworthy that the potential importance of tropospheric sulfur isotopic fractionations during OCS sinks. To date, sulfur isotopic fractionations were reported as −5 to 0 ‰ for reaction with OH radical (Schmidt et al., 2012), and for −2 to −4 ‰ for decomposition by soil microorganisms (Kamezaki et al., 2016; Ogawa et al., 2017), respectively. Sulfur isotopic fractionation for OCS by plant uptake, dominant OCS sink in troposphere (Berry et al., 2013), have not been determined, but the theoretical isotopic fractionation constant by plant uptake is −5.3 ‰ (Angert et al., 2019). Therefore, all sulfur isotopic

fractionation constants by OCS degradation are negative, indicating that the $\delta^{34}$S(OCS) values can be increased by OCS degradation in the troposphere. Because the main OCS sink is photosynthesis by plants, the $\delta^{34}$S(OCS) values in the atmosphere might be increased in the growing season for plants in April. However, because of the long lifetime of OCS, $\delta^{34}$S(OCS) values might not be sensitive to seasonal variation. Future studies must be conducted to determine the isotopic fractionation constants and observations of $\delta^{34}$S(OCS) values to estimate the dynamics of atmospheric $\delta^{34}$S(OCS) values in

the troposphere.

In addition to our observation of atmospheric $\delta^{34}$S(OCS) values with (10.5 ± 0.4) ‰ at Suzukakedai campus, Yokohama, Japan, recently $\delta^{34}$S(OCS) values with (13.4 ± 0.5) ‰ in August–October at Israel, (12.8 ± 0.5) ‰ in February–March in Israel, and (13.1 ± 0.7) ‰ in February–March at the Canary Islands, Spain were reported using GC/MC-ICP-MS method (Angert et al., 2019). These differences indicate that the atmospheric $\delta^{34}$S(OCS) values might not be homogeneous,

instead reflecting some geographic effects and/or potential difference for isotopic fractionations during sink processes. Given the higher influences of sulfur isotopic fractionations on $\delta^{34}$S(OCS) values during growing seasons, it is not likely to explain

lower atmospheric $\delta^{34}$S(OCS) values for Suzukakedai campus in April from those for Israel and Canary Islands observed in February–March. Rather, $\delta^{34}$S(OCS) values with (10.5 ± 0.4) ‰ at Suzukakedai campus might be more affected by anthropogenic OCS emission and/or less affected by oceanic OCS emissions compared to the samples collected in Israel/Canary Islands with higher $\delta^{34}$S(OCS) values. Potential anthropogenic OCS sources are Chinese emissions from rayon (yarn and staple) and coal (industry and residential), as pointed out by recent OCS source inventories (Zumkher et al., 2018). In fact, the OCS concentration in the vicinity of China is high based on satellite observation (Glatthor et al., 2015). Future study is necessary to observe spatial and temporal variation of $\delta^{34}$S(OCS) values to discuss the link between anthropogenic activity and OCS cycles.

In addition to tropospheric OCS sources, OCS have some potential as tracers of net ecosystem exchange into gross primary production (GPP) on land (Campbell et al., 2008). Based on our earlier experiments, to elucidate OCS in the troposphere and its relation to biochemical activity by plant and soil microorganisms, OCS sulfur isotope analysis provides a new tool to investigate soil OCS sinks in the troposphere. To date, we have determined the isotopic fractionation constants for OCS undergoing bacterial OCS degradation and its enzyme (Kamezaki et al., 2016; Ogawa et al., 10 2017). Similarly, additional studies that include specific examination of isotopic fractionation by plant uptake, another major sink of atmospheric OCS, are indispensable for distinguishing the respective OCS fluxes of soil and plants. By coupling isotopic fractionations by soil and plant with atmospheric observations of $\delta^{34}$S(OCS) values using our newly developed method, the atmospheric observations of $\delta^{34}$S(OCS) values are expected to help refine estimates of biological activities of plant and soil microorganisms and their respective contributions to OCS degradation in the troposphere.

**3.6 Comparison with other methods**

We here discuss the comparison of this sampling system coupled with GC/IR-MS and GC/MC-ICP-MS method (Said-Ahmad et al., 2017; Angert et al., 2019). The required sample amounts for our IR-MS system of over 6 nmol OCS. The overall precision value ($1\sigma$) for atmospheric $\delta^{34}$S(OCS) value is 0.4 ‰. By contrast, GC/MC-ICP-MS method (Said-Ahmad et al., 2017; Angert et al., 2019) has similar precision of 0.6 ‰, but only requires 20 pmol OCS. Consequently, IR-MS method requires a 300 times larger sample OCS than GC/MC-ICP-MS method. Therefore, our IR-MS method with a developed large volume air sampling system, has shortcomings for sample size and/or logistics for field campaigns. However, it is worth noting that benefits of our IR-MS method with its large volume air sampling system include its potential application of multi-isotope measurements of OCS by measuring $CO^+$ fragment ions for carbon and oxygen isotopes as well as $S^+$ fragment ions.

# 4 Summary

For this study, we developed a new OCS sampling and purification system. OCS is extracted from 500 L of ambient air was with a collection efficiency of almost over 95 % of OCS. The blank of the sampling and purification system was $(0.30 \pm 0.16)$ nmol and memory effects were negligible. By comparison with the previous used syringe injection (Hattori et al., 2015) we demonstrated that any potential isotopic fractionation during sampling and purification is negligible. The analytical repeatability values ($1\sigma$) for $\delta^{34}$S(OCS) value with more than 6 nmol for the commercial OCS samples and synthesized OCS samples were 0.2 ‰. We ascertained the $\delta^{34}$S(OCS) values for four atmospheric samples at Suzukakedai campus of Tokyo Institute of Technology located in Yokohama, Kanagawa, Japan. $\delta^{33}$S(OCS) were not reported because of a small overlapping signal on $m/z$: 33 in the ambient air samples. The OCS concentrations and $\delta^{34}$S(OCS) values respectively, were in the range of 447–520 pmol mol$^{-1}$ and 10.4–10.7 ‰. No clear diurnal variation in the $\delta^{34}$S(OCS) values was observed. Further modification of gas chromatographic techniques and/or data processing must be undertaken to measure $\delta^{33}$S(OCS) and $\Delta^{33}$S(OCS) values in future studies.

We earlier proposed a $\delta^{34}$S(OCS) value of $(4.9 \pm 0.3)$ ‰ for atmospheric OCS from measurements from a commercially available cylinder of compressed air (sample G in this study) (Hattori et al. 2015). Based on the four atmospheric samples taken in this study we revise this earlier value to $(10.5 \pm 0.4)$ ‰ being clearly distinct from the earlier value. The new $\delta^{34}$S(OCS) proposed here is in accordance with the $\delta^{34}$S(OCS) estimates of tropospheric and marine sources of OCS based on the OCS flux (Newman et al., 1991). Although OCS decomposition during preservation before the measurements was concerned, we found that no such OCS decomposition and isotopic fractionation have been observed for the modified adsorption tube made by Sulfinert-treated tube and valves and preservation at $-80$ °C at least within 90 days for OCS concentration and up to 14 days for $\delta^{34}$S(OCS) values.

Recently, Angert et al. (2019) reported the $\delta^{34}$S(OCS) value of ~13 ‰ in Israel or Canary Islands, and they suggested that the $\delta^{34}$S(OCS) value is homogeneous throughout the world. Although it is difficult to identify the reason for the difference of atmospheric $\delta^{34}$S(OCS) values between 10.5 ‰ in Japan and ~13 ‰ in Israel or Canary Islands, spatial and temporal variation of $\delta^{34}$S(OCS) values are expected to be a link between anthropogenic activities and OCS cycles.

**Acknowledgements**

We thank K. Yamada for his support of maintenance of GC/Q-MS. Thoughtful and constructive reviews by three referees led to significant improvements to the paper. This study was supported by JSPS KAKENHI (16H05884 (S.H.), 17J08979 (K.K.), 17H06105 (N.Y. and S.H.), from the Ministry of Education, Culture, Sports, Science and Technology (MEXT), Japan. For system development, this study is supported by research funds as a project formation support expenditure

"Internationalization of standards for quantification of biogeochemical process with innovated isotoplogue tracers" (N.Y.) from Tokyo Institute of Technology. E.B. acknowledges the Leibniz Association SAW funding for the project "Marine biological production, organic aerosol particles and marine clouds: a Process Chain (MarParCloud)", (SAW-2016-TROPOS-2).

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

**Table 1: OCS samples balanced with He and synthesized OCS sample of averages and standard deviations (1σ) for sulfur isotope ra[...] for OCS measured for this study (sampling/purification system with GC/IR-MS) and by conventional syringe injection system [...] GC/IR-MS as described by Hattori et al. (2015)**

| Sample | Sample type | Concentration | Supplier | DI-IR-MS (SF$_6$)[a] | | | | Syringe injection system with GC/IR-MS (S$^+$)[b] | | | | Sampling/Purification system with GC/IR-MS (S$^+$) | | | | Mo[...] S[...] sys[...] |
| --- | --- | --- | --- | --- | --- | --- | --- | --- | --- | --- | --- | --- | --- | --- | --- | --- |
| | | | | $n$ | $\delta^{33}S$ | $\delta^{34}S$ | $\Delta^{33}S$ | $n$ | $\delta^{33}S$ | $\delta^{34}S$ | $\Delta^{33}S$ | $n$ | $\delta^{33}S$ | $\delta^{34}S$ | $\Delta^{33}S$ | $n$ |
| | | | | | | (‰) | | | | (‰) | | | | (‰) | | |
| A | Commercial cylinder (Balanced with He) | 10.50% | - | 1 | 6.5[a] | 12.6[a] | 0.03[a] | 3 | 6.5 ± 0.2 | 12.6 ± 0.4 | 0.03 ± 0.1 | - | - | - | - | 6 |
| B | Commercial cylinder (Balanced with He) | 5.9 μmol mol$^{-1}$ | - | - | - | - | - | 3 | 7.0 ± 0.1[c] | 14.1 ± 0.2[c] | −0.2 ± 0.1[c] | 3 | 6.9 ± 0.4[d] | 13.8 ± 0.4[d] | −0.2 ± 0.4[d] | 3 |
| C | Synthesized (CO + S reaction) | 100% | Wako | - | - | - | - | 3 | −3.3 ± 0.1[d] | −6.3 ± 0.2[d] | −0.06 ± 0.1[d] | 3 | −2.8 ± 0.2[d] | −5.5 ± 0.4[d] | 0.03 ± 0.2[d] | - |
| D | Synthesized (CO + S reaction) | 100% | Sigma-Aldrich | - | - | - | - | 3 | 1.1 ± 0.2[d] | 2.4 ± 0.2[d] | −0.07 ± 0.1[d] | 3 | 1.5 ± 0.4[d] | 2.8 ± 0.7[d] | 0.08 ± 0.1[d] | - |
| E | Synthesized (CO + S reaction) | 100% | Mixture of Wako and Sigma-Aldrich | - | - | - | - | 3 | −1.5 ± 0.1[d] | −2.5 ± 0.2[d] | −0.2 ± 0.1[d] | 3 | −0.8 ± 0.6[d] | −1.9 ± 0.6[d] | −0.2 ± 0.3[d] | - |

[a] $\delta^{33}S$, $\delta^{33}S$, and $\Delta^{33}S$ values of SF$_6$ chemically converted from OCS in Sample A was corrected to values relative to the international standard (VCDT) nota[...] converted from IAEA-S-1(Ag$_2$S:$\delta^{33}S$ = −0.055 ‰ (Ono et al., 2007), $\delta^{34}S$ = −0.30 ‰ (Robinson, 1993) and $\Delta^{33}S$ = −0.100 ‰ (Ono et al., 2007)) and standa[...] of 25 times repeated measurement was 0.01 ‰.

[b] Developed system by Hattori et al. (2015)

[c] Corrected to values relative to the international standard (VCDT) notation by using sample A measured in this study

[d] Corrected to values relative to the international standard (VCDT) notation by using daily sample B injected from line developed by Hattori et al. (2015)

[e] Averaged and precision of $\delta^{34}S$(SF$_6$) value chemically converted from OCS in Sample A was corrected to values relative to the international standard (VC[...] SF$_6$ converted from IAEA-S-1(Ag$_2$S:$\delta^{33}S$ = −0.055 ‰ (Ono et al., 2007), $\delta^{34}S$ = −0.30 ‰ (Robinson, 1993) and $\Delta^{33}S$ = −0.100 ‰ (Ono et al., 2007)) (Hatto[...]

**Table 2: Sample information for compressed air in cylinders collected at Kawasaki, Japan**

| Sample | Concentration pmol mol$^{-1}$ | $\delta^{34}$S(OCS) ‰ | Experiments | Collecting date |
|---|---|---|---|---|
| F | 380 ± 15[a] | 11.7 ± 0.4 | Test of collection efficiency | 25 July 2017 |
| G | 168 ± 5[a] | 6.1 ± 0.4[d] | Determination of sulfur isotopic composition | 02 July 2012 |
| H | 200 ± 7[b] | - | Preservation test for OCS amount | 02 December 2017 |
| I | 371 ± 25[b,c] | 9.5 ± 0.4 | Preservation test for OCS amount | 26 October 2018 |
| J | 496 ± 30[c] | 9.3 ± 0.4 | Preservation test for OCS amount and $\delta^{34}$S(OCS) value | 01 December 2018 |
| K | 460 ± 29[c] | 10.4 ± 0.4 | Preservation test for OCS amount and $\delta^{34}$S(OCS) value | 26 December 2018 |

[a] Measured using Q-MS with pmol-level calibration curve

[b] Measured using Q-MS with nmol-level calibration curve after sampling

[c] Measured using IR-MS with calibration curve in Figure 4 after sampling with $1\sigma$ uncertainty of 6 %

[d] The previous $\delta^{34}$S(OCS) value measured by Hattori et al. (2015) was (4.9 ± 0.3) ‰.

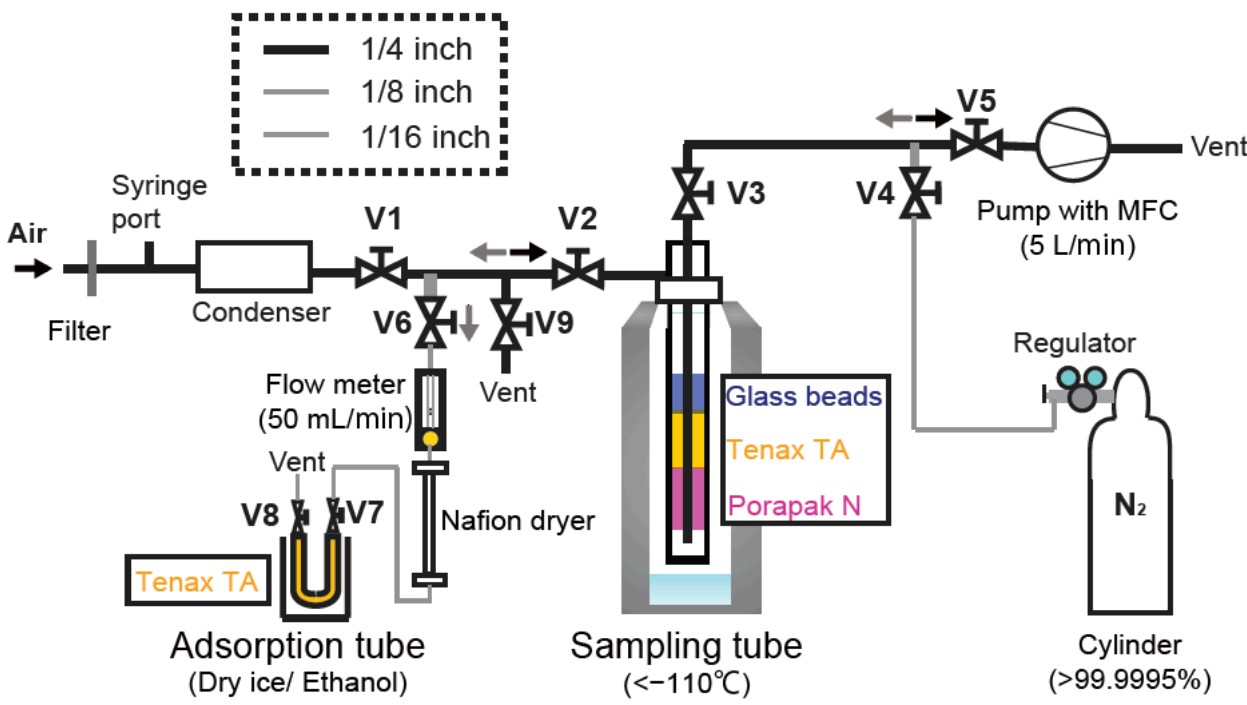

5 Figure 1: Schematic diagram of the OCS sampling system. System components: V, valve; pump, vacuum pump; MFC, mass flow controller.

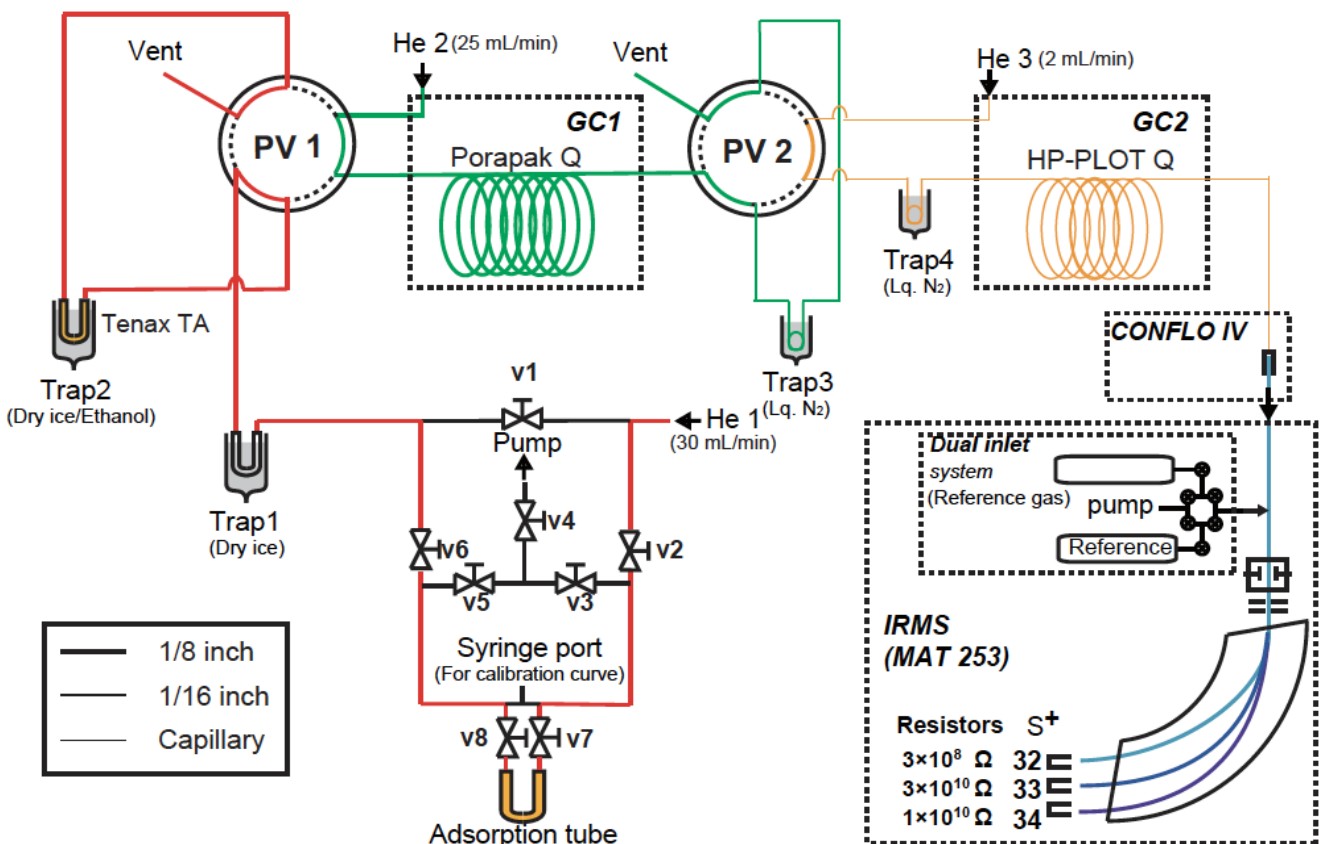

**Figure 2: Schematic diagram of the OCS purification system. System components: V, valve; pump, vacuum pump; MFC, mass flow controller.**

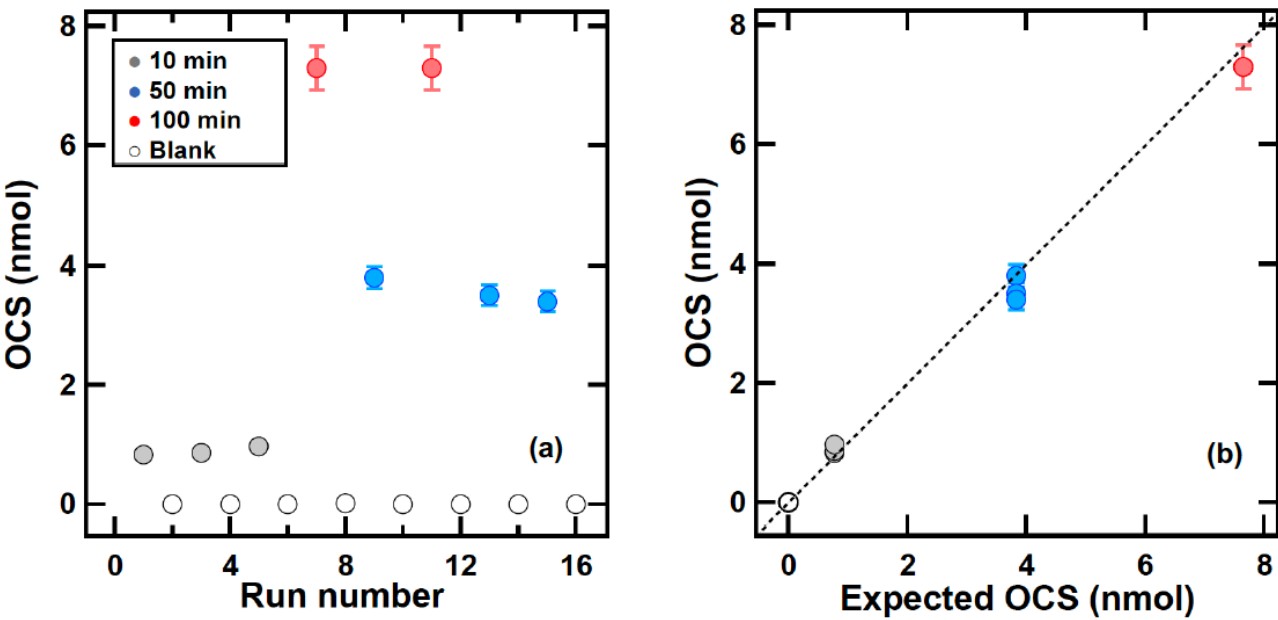

**Figure 3: OCS sampling using the sample F of (380 ± 15) pmol mol⁻¹ with different sampling times of blank (0 min), 10 min, 50 min, and 100 min. (a) Collected OCS amounts as a function of run numbers. (b) Observed OCS amounts and OCS amounts calculated using OCS concentration multiplied by the sampling time. Error bar shows ± 3 % based on the residual of measured OCS peak area and calibrated OCS peak area. Dotted line shows the slope of *x = y*.**

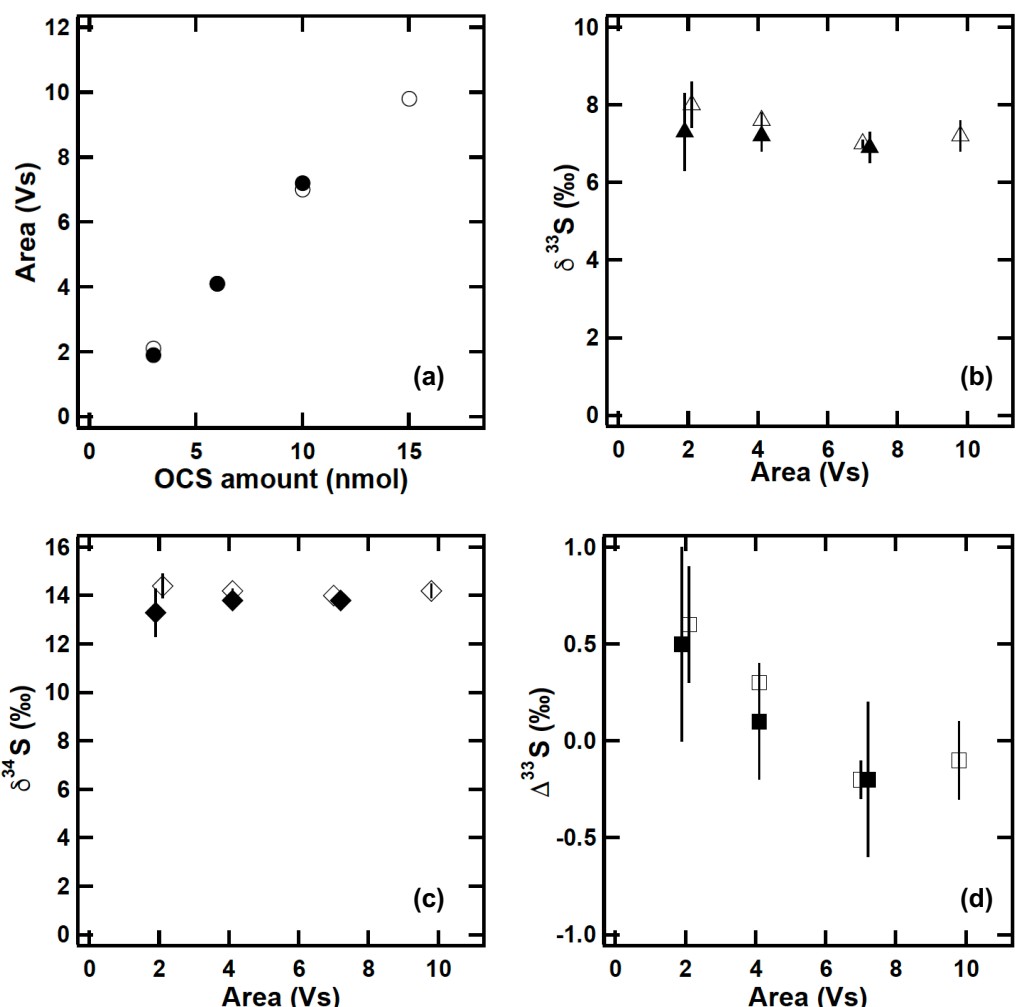

**Figure 4:** OCS amounts and sulfur isotope ratios of different amounts of OCS injections ascertained using the developed sampling/purification system and conventional syringe injection system (Hattori et al., 2015): (a) OCS amount; (b) $\delta^{33}S$; (c) $\delta^{34}S$; (d) $\Delta^{33}S$; closed symbols, sampling/purification system developed for this study; open

10 symbols, conventional syringe injection system. All sulfur isotope ratios are relative to VCDT. The error bars are $1\sigma$ of the measurements based on triplicated measurements.

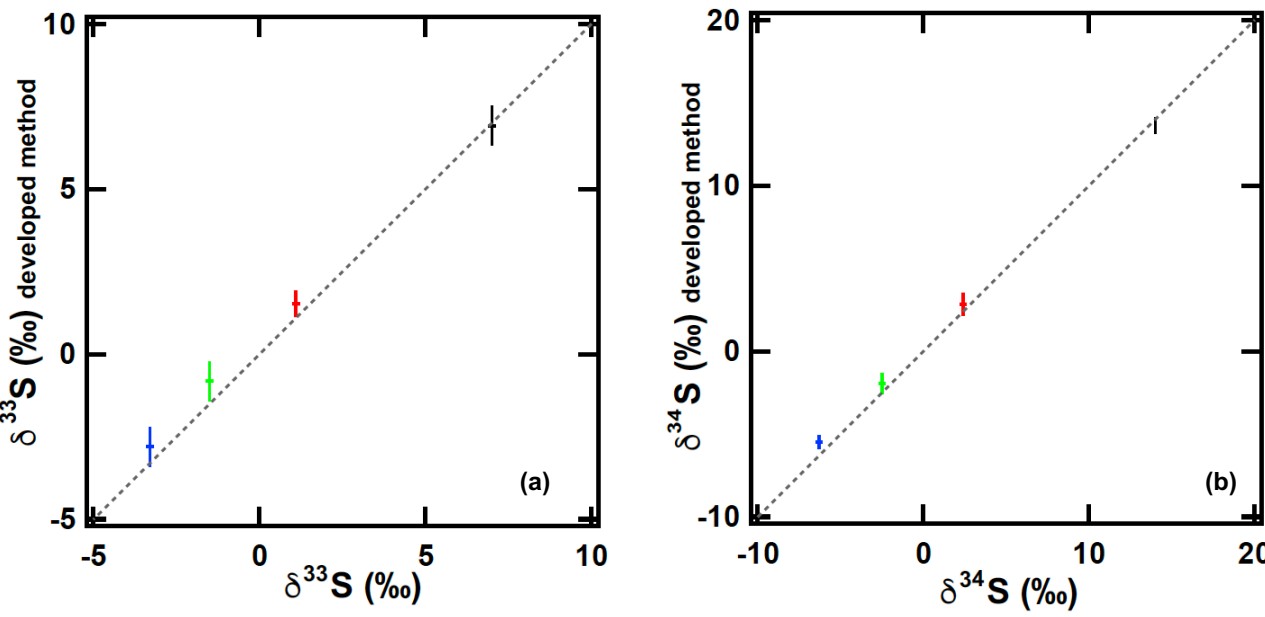

**Figure 5: Sulfur isotope ratios ((a) $\delta^{33}$S and (b) $\delta^{34}$S) ascertained from the developed sampling/purification system ($y$-axis) and conventional syringe injection system (Hattori et al., 2015) ($x$-axis). OCS sample amounts are 8 nmol. Different colors represent different samples: black, sample B; red, sample C; green, sample D; blue, sample E. Dotted line shows the slope $x = y$. The error bar is $1\sigma$ of each amount of triplicated OCS injection.**

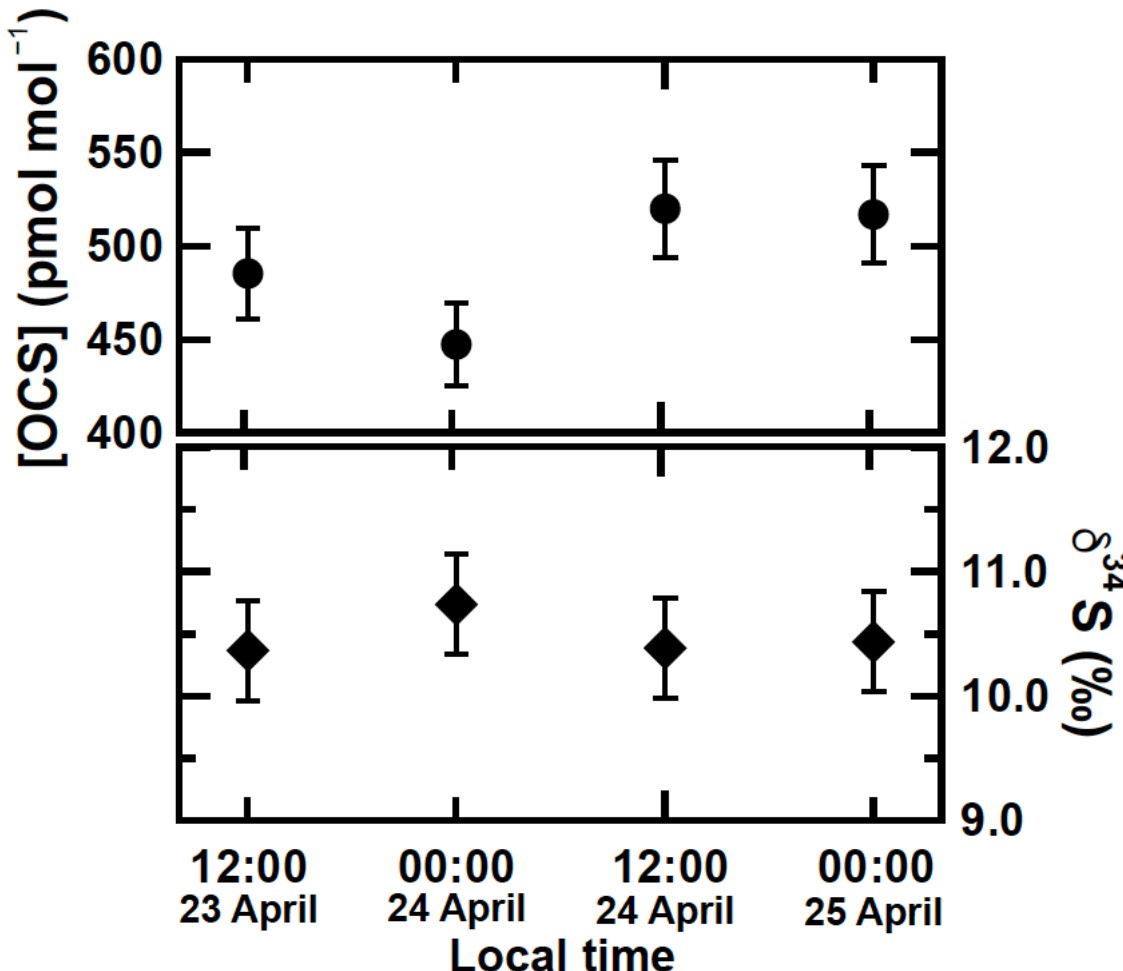

**Figure 6: OCS concentrations and $\delta^{34}$S(OCS) values for atmospheric samples collected at Suzukakedai campus of Tokyo Institute of Technology located in Yokohama, Japan. The error bar is 6 % for OCS concentration based on the precisions of syringe injection and flow rate of the diaphragm pump in the sampling system. The precision of $\delta^{34}$S is estimated from $1\sigma$ uncertainty of 0.4 ‰.**

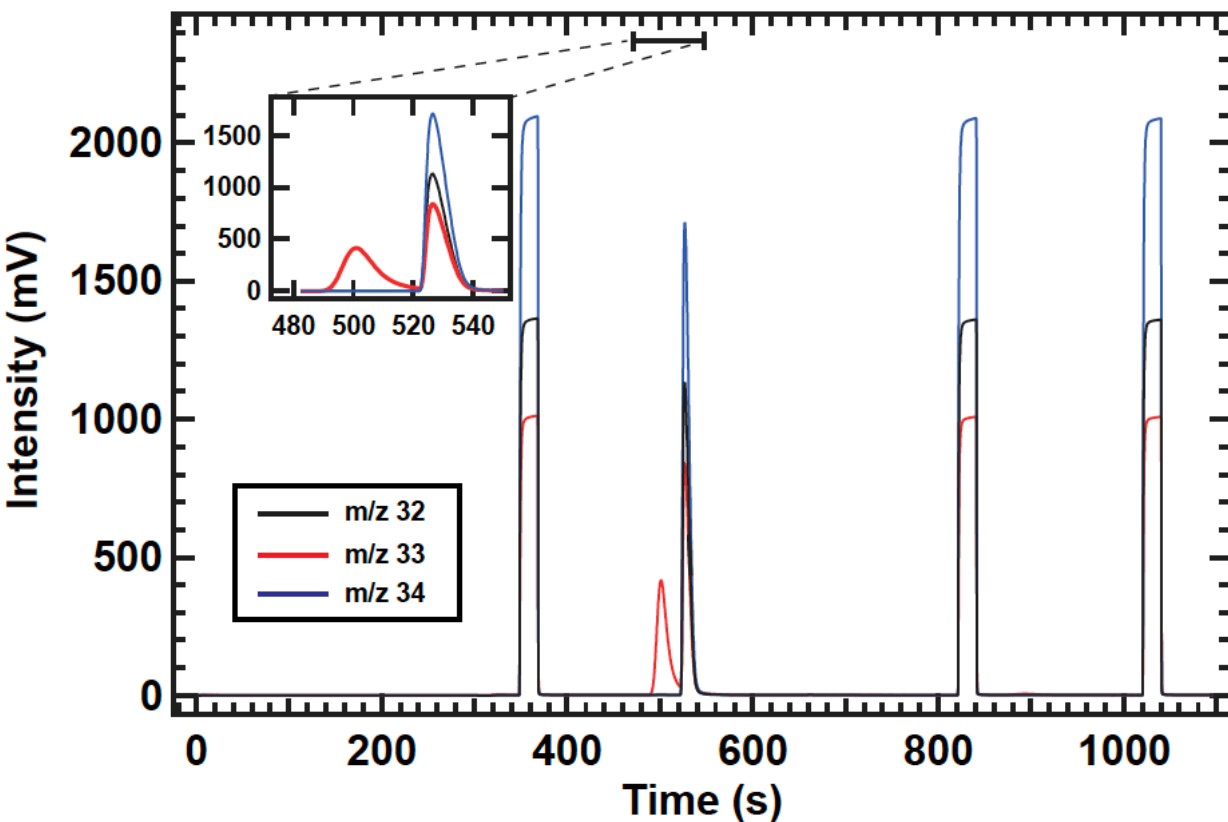

**Figure 7: IR-MS chromatogram of atmospheric samples collected at the Suzukakedai campus of the Tokyo Institute of Technology. Liquid N$_2$ removal from trap 4 occurred at 0 s in the purification system. Reference OCS was injected three times starting at 350 s, 825 s, and 1025 s for 20 s.**

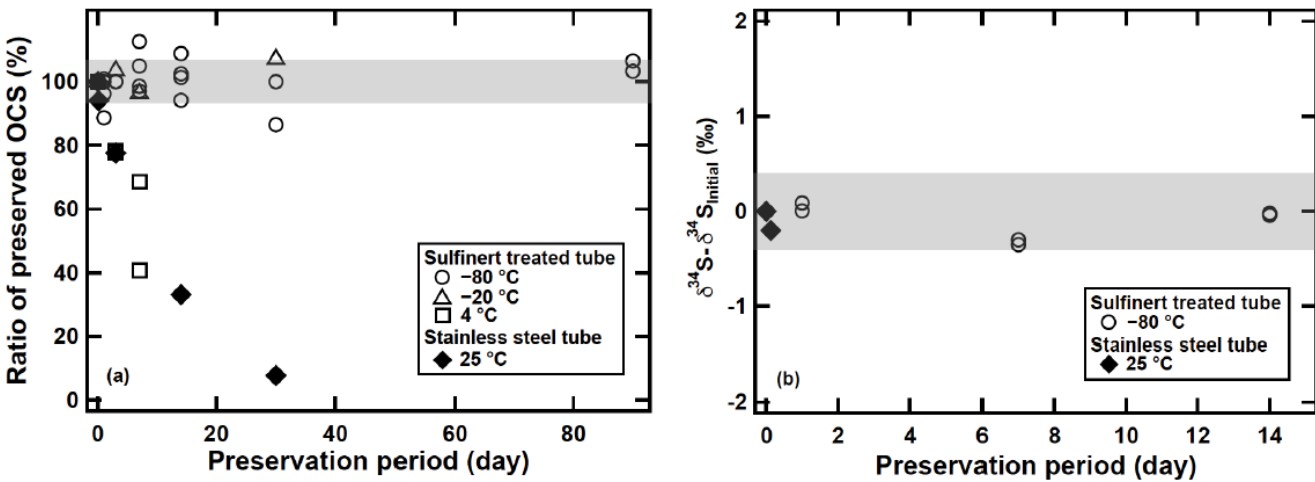

**Figure 8: (a) Changes in OCS concentrations preserved in the OCS storage test in adsorption tubes at different temperatures and tubes. (b) Changes of δ³⁴S(OCS) preserved in the OCS storage test. The shaded bar shows ± 6 % for OCS concentration and ± 0.4 ‰ for δ³⁴S(OCS) value based on the precisions of syringe injection and the flow rate of the diaphragm pump in the sampling system.**

