# Peer review of "Large volume air sample system for measuring $^{34}\text{S}/^{32}\text{S}$ isotope ratio of carbonyl sulfide"

_Atmospheric Measurement Techniques, 2018_

## Author Comment (AC1) · 11 Dec 2018

We recognized the paper ""Sulfur isotopes ratio of atmospheric carbonyl sulfide constrains its sources", which is currently in press in Scientific Reports. A preprint is available online since October 18th: https://www.essoar.org/doi/10.1002/essoar.10500051.3" mentioned by short comments after submission. We are happy to note about this findings in the last paragraph for the revised manuscript.
* * *

---

## Referee Comment (RC1) · Anonymous Referee #3 · 15 Dec 2018

The reviewed manuscript presents a method for collection and concentrating OCS (the most abundant atmospheric sulfur species) from air. The concentrated OCS sulfur isotopes are then measured by IRMS. Having such measurement method is an important contribution. The results seems robust, and the manuscript is usually well written.

The main problem I see regarding the suitability of this method for atmospheric sampling is related to samples preservation. So far, it is shown only for 7 days, which is not always enough to get a sample (i.e. from ship cruise or remote location) back to the lab and to analyze it. Also, no tests were done for preservation effect on isotopes. The authors themselves recognized this point as missing and wrote that "Further investigation of preservation of OCS concentrations and isotopic integrity during the storage of the adsorption tubes with respect to storage temperature and materials of the adsorption tube are currently underway."
I recommend that the results from these investigations will be included as part of this paper.

Detailed comments:

Page 1 (abstract), Line 17 – "7 inch tubes (1 cm^3)" – better to give both dimensions in cm.

Page 1 (abstract), Line 17 "preserving samples" – need to add "up to 7 days", or to whatever the new investigations show.

Page 1 (abstract), line 20 – should be "lack of diurnal variations".

Page 1 (abstract) line 20 and Page 11 line 6 – It appears here as this method enables measurements also of 33S in air samples. In fact, this has worked out only for standards, and an interference prevented measuring this in air. Please correct.

Page 1 (abstract) – Portability is reported here based on the sample tubes. But the size and weight of the sampling system is also a main issue. Please report this, at least in the method section.

Page 3, line 30 – Please explain the need for both glass-beads, Tenax TA, and Porapak N. On which of those is the OCS trapped? All of them?

Page 4, line 7 – "less than -110C". Can you be more specific? It can be important for someone trying to use this method. Also, if you get as low as liquid N2 temperature, than some O2 (having higher boiling point) will be liquefied.

Page 6, line 5 – Rephrase. It should be explained in a different sentence that standard A is not pure OCS because of …

Page 7, line 11 – Refer to figure 3a.

Page 7, line 23-24 – Not clear. Please rewrite.

Page 8, Line 16. This is only true if the blanks always have atmospheric values. Please consider how it will affect the uncertainty in atmospheric measurements, if the blanks are in the expected range of OCS sources.

Page 9, line 3 – The variability in concertation reported by Montzka (2007) is seasonal, not diurnal. Any possible explanation for the diurnal variability, and how it is related to what reported for other sites?

Page 9, line 12 – Is this 1 permil change (not this small) is also accompanied by a change in concertation? Can this be related to the blank problem? Please explain in the text.

Page 9, line 19 – What do you mean by "reasonable signals"

Page 9, line 20 – "We earlier discussed", where? In previous papers by the same group? I tried to read this in Hattori et al. (2015), but it is not clear to me. Please explain in more detail here how this isotopic balance was done.

Also, if both the earlier value and the new value agree with the SSA value, then this is not a strong constraint for atmospheric OCS d34S.

Page 10, Line 3 – Please report the magnitude of these effects, and estimate how this should influence the atmospheric OCS.

Page 10, line 29 – These "further investigations" seem critical to establish the method. Are there new results since the manuscript was first submitted?

Page 11, line 26 – Please be more specific. Instead of "some shortcomings in terms of sample amount" write down the ratio of sample amounts in the two methods.

Page 11 - The conclusion section mentions for the first time another method (by Said-Ahmad, 2017). The mentioning of other existing methods should be done in the introduction, and comparison with other methods should be in the discussion, and include advantages and disadvantages. Similarly, writing in the last lines of the paper that the current method has the advantage of enabling carbon and oxygen isotopes measurements seems out of place, since this is not demonstrated in the current manuscript. However, this can be mentioned in the discussion.

The discussion should also refer to the new paper by Angert et al., mentioned in the online discussion. Are the atmospheric values reported by the two papers identical, considering all experimental uncertainties? If not, is this a methods issue, or a real geographic effect?

Figure 3a – Maybe better to show versus sampling time (and not run number). Also the 10min is black in the legend and gray in the figure.

Figure 8 – Better to start the y-axis at 80%. Also, need to show longer preservation periods.

---

## Referee Comment (RC2) · Anonymous Referee #2 · 21 Dec 2018

This manuscript attempts to address a current technical challenge within the atmospheric S and to some extent C cycle research community. It is clear that quantifying the movement of sulphur between the ocean, atmosphere and biosphere and detailing potential interactions and transformations is highly desirable but unfortunately it is still an extremely difficult task to conduct. Carbonyl sulphide is the most abundant sulphur containing gas in the atmosphere and currently there is much debate about its current sources and sinks. Within this framework the use of COS isotopic tracers might provide insights on this problem and help close the COS mass budget. As pointed out in the manuscript this not only holds interest for the communities working on the S cycle and its interactions with climate through chemical reactions in the troposphere and stratosphere but also those working on the carbon cycle, as COS is a close analog of CO2 and as such moves between the atmosphere and the biosphere alongside CO2 during various steps of the photosynthetic pathway.

Consequently the motivation to study COS isotopes is growing and there is a need to overcome a number of technical challenges that limit the application of these tracers to help address the scientific questions introduced above. The first problem is that COS is a trace gas (atmospheric concentration around 500parts per trillion) with an extremely low abundance of the rare isotope species thus large sample volumes are required for Isotope Ratio Mass Spectrometry (around 500L providing ∼10nmol of COS, in the case of the present study). Secondly, COS is an extremely reactive gas that can be hydrolysed and/or produced rapidly from many surfaces and materials commonly used in gas exchange techniques.

The current study presents a large volume sampling system adapted from a previous system designed for the measurement of volatile halocarbons and hydrocarbons by one of the co-authors of the present study (Bahlmann et al., 2011). This trapping system seems to be capable of tackling the first problem described above by trapping high volumes of air over a relatively short period of time (∼100 minutes) and concentrating the air sample in a cryoshipper volume cooled previously with liquid N2. Details regarding the temperature of the cryoshipper volume are a little vague stated as less than -110oC. This could be a bit more specific as pointed out by one of the reviewers as other gases may condense at lower temperatures.

The second problem of sample contamination during sampling and sample storage seems to still be an issue that would prevent the routine use of this sampling approach. For instance the authors purge the sampling tubes with high-purity Helium for 12H before sampling and require high temperatures starting at 160oC and reaching 330oC to condition the material. Can this be done in the lab beforehand on a batch of sampling tubes and does the surface remain inert thereafter, if so for how long? Is this step necessary when switching to Sulfinert valves and tubing? I am also curious to know

how long roughly it takes to process one sample from the pre-conditioning stage to the final analysis of the purified sample on the IRMS?

It was useful to see the change in COS concentration during the preservation period. However, if I understand correctly this is not the experimental set-up for the rest of the data presented in the paper. Can you also plot the preservation data timeline for the non sulfinert experimental set-up please?

Reviewing this as a method paper I feel there were a number of details missing or communicated a bit ambiguously. I am not sure it would be so easy to reproduce this methodology as a result. In particular section 2.1 was rather vague in several places, especially when it came to details of the calibration cylinders being used. Nowhere is the material of any of the calibration cylinders stated. This is not trivial as it is well known that COS is highly reactive and the use of stainless steel or aluminium cylinders for storage of COS standards will cause a drift in the COS concentrations over time and I would assume the isotopic composition too. For example, you state that the calibration cylinders F and G are much lower than that of atmosphere (e.g. G = 160 pmol mol-1), despite being sampled from the atmosphere that should be in the range of 350-500 pmol mol-1. I can only assume that contamination has occurred in the bottles during storage or a scrubber has been used whilst filling the tanks. If I understand these cylinders were filled in 2012 and assumed to represent the global background atmospheric composition. I am keen to know when the calibration curve described in 2.4.1 was actually completed a few days, months or years (2012) before the use of the calibration gases to validate the field measurements for COS concentration and $\delta 34S$? Overall calibration cylinders should be monitored closely over time when filled to see if the cylinder has issues and thereafter regularly checked for drift. It is also recommended to use Sulfinert cylinders or ACULIFE cylinders for COS. Can you confirm whether your calibration cylinders have special wall treatments to minimize contamination of your atmospheric COS gas?

I also think it would be worth discussing a little the caveats surrounding the fragmentation IRMS approach for example the potential for O2 contamination and consequences for the $\delta 34S$ and $\Delta 33S$ analysis/interpretation.

Finally I think section 3.5 breaks the flow of the paper at the end and think it should rather become section 3.3. This would allow the paper to move out of the technical discussion into the scientific discussion and conclude more naturally. I also feel that the first paragraph of the conclusion is repetition of the results and should be removed. I also agree with the other reviewer that the last paragraph introducing the carbon and oxygen isotopes should not suddenly appear here a bit out of the blue, their measurement is not trivial.

Other details

The application of parentheses throughout this paper needs correcting, just because you quote a number does not mean it should be wrapped in brackets especially if it is integral to the sentence structure e.g. Page 1 line 26 and 27 highlights the type of problem that pops up throughout the manuscript. Please double check all parentheses are appropriate.

Page 1 line 26-27 How can you prove which value is correct? They both seem to be within the range of values for S sources. I would also try to clarify this argument/sentence a bit better. Page 2 Line 8 remove "net ecosystem exchange into" from this sentence. Page 3 Line 11 state the purity of the Helium in % Page 3 Line 12 can you provide more details of the exact S compounds used Page 3 Line 16 what type of commercial cylinders provide details and how often they were measured. Can you also provide details of the compressor used to fill the bottles and any filters that were used on the compressor? Was the air dried before filling the cylinders? Page 3 Line 20 this is a super vague statement can you provide details of how you checked for stability. Page 4 Line 3-4 can you explain how the heating was made and where was this step made in the lab or outside? Page 4 Line 15 removed using what? Page 4 Line 27-28 is the tubing flexible i.e. in a coil or rigid? Page 4 line 29 how is the heating

achieved? Page 4 line 30 why use steel and not glass. I guess this was later switched to a sulfinert tube? Page 5 line 8 this should be -72oC Page 5 Line 11 where did this injected gas come from? Page 5 lines 22-25 not written very clearly please improve Page 5 line 25 how many replicates were analysed to obtain the precision? Page 6 line 11 size of capillary tube? Page 6 line 9 spec of the diaphragm pump type, flow, pressure? Page 6 line 16 tank ID please Page 7 line 18-20 this sentence does not make sense to me Page 7 line 21-22 ambiguous sentence Page 7 line 26 how? And what does slowly mean? Page 8 line 15 could you explain these OCS blanks please Page 9 line 11 check the units here please and correct Page 9 line 14 hydrolysis requires water were your tanks dry? Page 9 line 28-30 this sentence needs rewritten as it does not make sense to me Page 10 generally there is little statistical testing reported in this paper in general Page 10 this section would really benefit from more results using the sulfinert equipment.

---

## Referee Comment (RC3) · Anonymous Referee #1 · 2 Jan 2019

To avoid unnecessary duplication, I will restrict my comments to aspects not mentioned by referees #2 and #3 yet.

1. The "large volume" aspect needs to be specified – clearly it refers to large volumes of air. Given that only the $^{34}S/^{32}S$ ratio could be analysed successfully, the title of the paper should be changed to "Large volume sampling system for measuring the $^{34}S/^{32}S$ isotope ratio of atmospheric carbonyl sulfide", or something along these lines.
2. Section 2.4.1 should be renumbered 2.4 and renamed "Determination of the OCS concentration".
3. Section 2.4.2 should be renumbered 2.5 and renamed "Determination of the sulfur isotopic composition of OCS".
4. Table 2: You should include the results for the sulfur isotope deltas of samples A, F, G and H in the left hand column of this table, for ease of reference. Possibly, you could also present them in a separate table, given that sample G was analysed by Hattori et al. (2015) already, but gave a different result.
5. Table 2: Given that samples B, C and D all seem to have drifted with respect to the reference sample A, how did you ensure that the composition of sample A itself has not changed compared to the previous 2015 paper?
6. 5/31: One could hypothesise that samples F, G and H all started out at the same initial OCS mole fraction and isotope ratio. Adopting this hypothesis, could you please include a plot of their isotope deltas vs. the natural logarithm of the "residual" OCS fraction (i.e. a Rayleigh fractionation plot) to check whether the apparent OCS loss in the cylinders follows a common fractionation constant $\varepsilon$?
7. 7/26: Please describe in detail how you introduced these aliquots of sample B?
8. 8/27: The $m/z$ 33 interference could also be due to $NF^+$ (e.g. from $NF_3$).
9. 11/25: Please state the precision achieved for OCS analysis in this earlier paper.
10. 22/9: The precision achieved for sample B is not meaningful for these air samples. Please replace the error bars with a more suitable estimate of the precision for an actual air sample.
11. Referee #2 commented on the use of parentheses in your manuscript. The notation "$(x \pm s_x)$ ‰" (and similar) follows in fact international guidelines on the SI such as NIST Special Publication 811 2008 Edition "Guide for the Use of the International System of Units (SI)" and the IUPAC Green Book, 3rd edition, p. 151 (section 8.1, example 2; http://www.iupac.org/fileadmin/user_upload/publications/e-resources/ONLINE-IUPAC-GB3-2ndPrinting-Online-Sep2012.pdf. As the journal advocates the use of the SI, no change is necessary.

**Technical corrections**
- 2/4: Brühl et al.
- 2/11: S equivalents
- 2/17: O($^3$P) – spin states are not written in italics
- 3/24: compartments

- 3/25 & various occurrences elsewhere: Sulfinert
- 5/30: Samples F, G and H
- 7/23: Add full-stop after system and start new sentence "We sequentially ..."
- 7/25: dependence
- 7/31 & 32: Replace full stop after $\sigma$ with "uncertainty: ".
- 8/26 to 8/28: Remove colon (:) after $m/z$ (e.g. $m/z$ 32)
- 9/5: USA
- 11/5: proofed -> showed

---

## Author Comment (AC2) · 28 Jan 2019

Reply to RC1 from referee #3

Blue: Referee comments Black: Our comments Red: The sentences in our manuscript

The reviewed manuscript presents a method for collection and concentrating OCS (the most abundant atmospheric sulfur species) from air. The concentrated OCS sulfur isotopes are then measured by IRMS. Having such measurement method is an important contribution. The results seems robust, and the manuscript is usually well written.

**Reply:** Thank you very much for your comments.

The main problem I see regarding the suitability of this method for atmospheric sampling is related to samples preservation. So far, it is shown only for 7 days, which is not always enough to get a sample (i.e. from ship cruise or remote location) back to the lab and to analyze it. Also, no tests were done for preservation effect on isotopes. The authors themselves recognized this point as missing and wrote that "Further investigation of preservation of OCS concentrations and isotopic integrity during the storage of the adsorption tubes with respect to storage temperature and materials of the adsorption tube are currently underway. I recommend that the results from these investigations will be included as part of this paper.

**Reply:** We agree that the preservation period of OCS in adsorption tubes is important. During the process of the review, we investigated preservation of OCS in adsorption tubes at 25 °C, 4 °C, -20 °C, and -80 °C. At -80 °C, we found that the  $\delta^{34}S(OCS)$  value is not changed at least up to 14 days. The OCS concentrations are preserved up to 90 days. Therefore, we added data reflecting these experimental results to our manuscript.

**Detailed comments:**

Page 1 (abstract), Line 17 - "7 inch tubes  $(1 \text{ cm}^{-3})"$  – better to give both dimensions in cm.

**Reply:** We apologize for inappropriate expressions because it is difficult to identify what 7 inches was referring to. We replaced "inch" units with "cm".

Action: We described the sentence in page 1 line 18 as follows: adsorption tubes (1/4 inch (0.64 cm) outer diameter, 17.5 cm length, approx. 1.4 cm3 volume).

Page 1 (abstract), Line 17 "preserving samples" – need to add "up to 7 days", or to whatever the new investigations show.

**Reply:** As we replied above, we have confirmed the preservation of OCS amount and its  $\delta^{34}S(OCS)$  values in Sulfinert-treated adsorption tubes at -80 °C up to 90 days and 14 days, respectively.

Action: We added "the OCS amount and  $\delta^{34}S(OCS)$  values at -80 °C, respectively, for up to 90 days and 14 days," here.

Page 1 (abstract), line 20 – should be "lack of diurnal variations". **Reply:** We replaced "No significant diurnal variation" with "lack of diurnal variation" here.

Page 1 (abstract) line 20 and Page 11 line 6 - It appears here as this method enables measurements also of 33S in air samples. In fact, this has worked out only for standards, and an interference prevented measuring this in air. Please correct.

**Reply:** As you have pointed out, the 33S for atmospheric OCS was not measured. Therefore, we deleted the precision of  $\delta^{33}$ S and  $\Delta^{33}$ S values for atmospheric OCS in the *Abstract* and *Conclusion* (summary).

Page 1 (abstract) – Portability is reported here based on the sample tubes. But the size and weight of the sampling system is also a main issue. Please report this, at least in the method section.

**Reply:** The sampling system size and weight are 50 cm  $\times$  50 cm  $\times$  50 cm (width  $\times$  height  $\times$  depth), and 4 kg except for a dewar (37 cm outer diameter, 66 cm height and 11 kg weight). For the field campaign, the system was disassembled. We carried the parts of the system in two containers 40 cm  $\times$  30 cm  $\times$  20 cm (width  $\times$  height  $\times$  depth) except for the dewar. Then, we were able to assemble the sampling system on site and use the system. Therefore, we can carry our sampling system to field campaigns. We added this information to our manuscript.

Action: We added sampling system size and weight information and handles for the field campaign to our revised manuscript on 1st paragraph of section 2.2.

Page 3, line 30 – Please explain the need for both glass-beads, Tenax TA, and Porapak N. On which of those is the OCS trapped? All of them?

**Reply:** We are sorry that we did not specify the necessity of each adsorbent in the sampling tubes. The sampling tube design is based upon the description by Bahlmann et al. (2011). According to Bahlman et al. (2011), glass beads trap the remaining water vapor from the sampled air and prevent adsorption of the water vapor on the Tenax TA and Porapak N and increase the temperature exchange between walls of the cryotrap and the sampled air. The Tenax TA and Porapak N can be used for trapping volatile organic compounds. We assume that OCS as our purpose is supposed to be sampled on the Tenax TA and/or Porapak N, but most of OCS might be trapped Tenax TA. Although some components might not be necessary for OCS collections, it works well for OCS sampling. Therefore, we do not plan to modify these in this system.

Action: In the revised manuscript, we described the following on  $2^{nd}$  paragraph of section 2.2: "We developed this sampling tube according to Bahlmann et al. (2011). Detailed functions of respective components are described therein. Briefly, the glass bead traps the remaining water vapor from the sampled air and prevents water vapor adsorption on the Tenax TA and Porapak N. The glass bead further increases the temperature exchange between the cryotrap walls and the sampled air. The Tenax TA and Porapak N can be used for trapping volatile organic compounds. We assume that OCS is sampled on the Tenax TA and Porapak N, but most of OCS might be trapped on Tenax TA. Although some components might not be necessary for OCS collections, up to this point, it is working well for OCS sampling."

Page 4, line 7 – "less than -110C". Can you be more specific? It can be important for someone trying to use this method. Also, if you get as low as liquid  $N_2$  temperature, than some  $O_2$  (having higher boiling point) will be liquefied.

**Reply:** We are sorry for the ambiguous sentence. Although it is difficult to keep the temperature constant using vapor of liquid N2, we confirmed that the temperature in the dewar was at least -140 °C to -110 °C for 3 hr. Because the sampling tube temperature is over -140 °C, O2 is not trapped in the sampling tube during sampling. **Action:** We added the temperature during sampling 4th paragraph of section 2.2 (page 5 line 7) as follows: at temperatures of -140 °C.

Page 6, line 5 – Rephrase. It should be explained in a different sentence that standard A is not pure OCS because of ...

Reply: Thank you for your comments.

Action: We separated the sentence in 1st paragraph of section 2.5 as follows: Reference OCS of sample A was purified with liquid N2 (-196 °C) and then introduced via a conventional dual inlet system. Pure OCS is not commercially available in Japan because of its toxicity (Hattori et al. 2015).

Page 7, line 11 – Refer to figure 3a. **Reply:** Corrected accordingly.

Page 7, line 23-24 – Not clear. Please rewrite.

**Reply:** We apologize for the unclear sentence.

Action: We rewrote the text to clarify the sentences ( $1^{st}$  paragraph at section 3.2) as follows: In the developed system, the possibility exists that OCS is lost by passing OCS through GC1. Also, because the flow rate of approx. 50 mL / min was lower than the flow rate of approx. 200 mL / min reported by Hattori et al. (2015), the possibility exists that OCS was lost by Trap 1. Therefore, to assess these possibilities, the following test was conducted. Firstly, 5 nmol of OCS was injected to a system consisting of Trap 2, GC2, and Trap 4 and measured as true value. Then, the same amount of OCS was introduced into the developed purification system and the amount of OCS obtained was compared to true value.

Page 8, Line 16. This is only true if the blanks always have atmospheric values. Please consider how it will affect the uncerainty in atmospheric measurements, if the blanks are in the expected range of OCS sources.

**Reply:** Thank you for your suggestion. We estimated the contamination effect for accuracy and precision of  $\delta^{34}S(OCS)$  values when the 5 % of contaminated OCS ranging  $\delta^{34}S(OCS)$  value from 3 to 18 ‰ as follows. 5 % of OCS contamination change the accuracy of  $\delta^{34}S(OCS)$  value with  $\pm$  0.3 ‰. The precision of our repeated measurement is  $\pm$  0.2 ‰. Then the overall precision of measurement is  $\pm$  0.4 ‰. Additionally, the standard deviation of four atmospheric samples we observed are  $\pm$  0.2 ‰. Therefore, the  $\delta^{34}S(OCS)$  value for atmospheric OCS at Suzukakedai campus was (10.5  $\pm$  0.4) ‰. We modified precision and Figure 6 in our manuscript.

Action: We added some text to 2nd paragraph of section 3.2 as follows: When considering  $(0.30 \pm 0.16)$  nmol OCS (i.e. approx. 4 % for 8 nmol OCS samples) with  $\delta^{34}$ S of 3–18 ‰ covering reported  $\delta^{34}$ S range of OCS sources (Newman et al., 1990), the accuracy of the  $\delta^{34}$ S(OCS) can be shifted –0.3 to +0.3 ‰. Because the precision of 1 $\sigma$  uncertainty is 0.2 ‰, the overall precision values (1 $\sigma$ ) for  $\delta^{34}$ S of this sampling/purification system were estimated as 0.4 ‰. Additionally, we modified the error in Figure 6.

Page 9, line 3 – The variability in concertation reported by Montzka (2007) is seasonal, not diurnal. Any possible explanation for the diurnal variability, and how it is related to what reported for other sites?

**Reply:** As you have pointed out, Montzka et al. (2007) did not observe diurnal variation of OCS concentration. For the diurnal variation of OCS concentration, we expected that the OCS concentrations are low at 12:00 compared to 00:00 because of plant uptake in our observation. However, we did not observe the trend. Berkelhammer et al. (2014) reported diurnal variation for OCS concentrations in USA with the lowest at 8:00 and the highest at 16:00 with 80 pmol mol-1 changes in a day. The differences of OCS concentrations for four atmospheric samples were smaller than 80 pmol mol-1. The observed  $\delta^{34}$ S(OCS) values of four atmospheric samples were in the range of 10.4–10.7 ‰ (Figure 6b) and averaged (10.5 ± 0.4) ‰, and  $\delta^{34}$ S(OCS) values also showed no clear diurnal difference (*p*-value = 0.29) (Figure 6b). Given the diurnal OCS variations, future study is clearly necessary to test whether or not  $\delta^{34}$ S(OCS) values have diurnal variations by comparing  $\delta^{34}$ S(OCS) values for the highest OCS concentration at 8:00 and the lowest OCS concentration at 16:00. We added related discussion of the matter to the revised manuscript (3rd paragraph of section 3.3).

**Reference**

Berkelhammer, M., Asaf, D. Still, C., Montzka, S., Noone, D., Gupta, M., Provencal, R., Chen, H. and Yakir, D.: Constraining surface carbon fluxes using in situ measurements of carbonyl sulfide and carbon dioxide, Global Biogeochem. Cycles, 28, 161–179, 2014. doi:10.1002/2013GB004644.

**Page 9, line 12 - Is this 1 permit change (not this small) is also accompanied by a change in concertation? Can this be related to the blank problem? Please explain in the text.**

**Reply:** Yes, the 1 ‰ change might be caused by both isotopic fractionation for OCS decomposition and blank effect. When the contaminated OCS with  $\delta^{34}S(OCS)$  value of over 17 ‰ are considered, the  $\delta^{34}S(OCS)$  value can be increased by 1.2 ‰ in sample G. However, such a high  $\delta^{34}S(OCS)$  value in the blank is not reasonable because the contaminated OCS coming only from the ocean is not likely. Indeed, because the atmospheric  $\delta^{34}S(OCS)$  value observed in this study was 10.5 ‰, and OCS concentration in sample G was lower than atmospheric OCS concentration, the increased  $\delta^{34}S(OCS)$  value is expected to be affected by isotopic fractionation during OCS degradation in the cylinder.

Action: We added some text to the revised manuscript as follows: It is possible to explain this 1.2 % increase for  $\delta^{34}S(OCS)$  value for a case in which the contaminated OCS has  $\delta^{34}S(OCS)$  value with over 17 %. However, such a high  $\delta^{34}S(OCS)$  value from contamination requires a situation in which the contaminated OCS come only from the ocean, which is not likely. Because the atmospheric  $\delta^{34}S(OCS)$  values in this study were (10.5 ± 0.4) % and higher than that for sample G, the increased  $\delta^{34}S(OCS)$  values are expected to be affected by isotopic fractionation during OCS degradation in the cylinder and not by contamination.

**Page 9, line 19 – What do you mean by "reasonable signals"**

**Reply:** We apologize for the lack of clarity. We intended the words "reasonable signals" as a representative value when compared to  $\delta^{34}S(OCS)$  value of  $(4.9 \pm 0.3)$  ‰ reported in Hattori et al. (2015) because the atmospheric  $\delta^{34}S(OCS)$  value of 10.5 ‰ is close to 11 ‰ as estimated based on the mass balance of OCS source from land and oceans. Additionally, because the OCS of compressed air in the cylinder can be decomposed and the  $\delta^{34}S(OCS)$  value may be changed in the cylinder, we do not know if the OCS in the cylinder represents atmospheric OCS. If we consider the  $\delta^{34}S(OCS)$  value of 13‰ in Israel and Canary Islands reported by Angert et al. (2018) during process of review, the  $\delta^{34}S(OCS)$  values might not be homogeneous in the world. Action: We deleted these words from our manuscript.

**Page 9, line 20 – "We earlier discussed", where? In previous papers by the same group? I tried to read this in Hattori et al. (2015), but it is not clear to me. Please explain in more detail here how this isotopic balance was done. Also, if both the earlier value and the new value agree with the SSA value, then this is not a strong constraint for atmospheric OCS $\delta^{34}$ S.**

**Reply:** We are sorry for an inadequate explanation. First, "We earlier discussed" refers to the discussion described in Hattori et al. (2015) and by Leung et al. (2002). As you have pointed out, both  $\delta^{34}S(OCS)$  value of  $(4.9 \pm 0.3)$  ‰ and  $(10.5 \pm 0.4)$  ‰ agree with the  $\delta^{34}S(OCS)$  value of SSA value, indicating that it is not a strong constraint of  $\delta^{34}S(OCS)$  value for atmospheric OCS. However, we earlier hypothesized the  $\delta^{34}S(OCS)$  value of  $(4.9 \pm 0.3)$  ‰ reported by Hattori et al. (2015) as a global signal, but it would be not correct. Therefore, we inferred the importance of showing that the new observed  $\delta^{34}S(OCS)$  value is suitable as a sulfur source of SSA, but it is still similar to a discussion put forth by Schmidt et al. (2013), who hypothesized a  $\delta^{34}S(OCS)$  value of 11 ‰ according to Newman et al. (1990). Because the discussion has already been reported and because the  $\delta^{34}S(OCS)$  value is close to 11 ‰, we deleted these discussions and instead used a more detailed description in the revised manuscript.

Action: We delete these discussions in favor of more detailed description in the revised manuscript.

Page 10, Line 3 – Please report the magnitude of these effects, and estimate how this should influence the atmospheric OCS.

**Reply:** First, sulfur isotopic fractionations of OCS for the troposphere was estimated as -5 to 0 ‰ for reaction with OH radical (Schmidt et al., 2012), -2 to -4 ‰ for decomposition by soil microorganisms (Kamezaki et al., 2016; Ogawa et al., 2017) and -5.3 ‰ for plant uptake (Angert et al., 2018). We added these values to our manuscript. We added Angert et al. (2018) as a new reference.

Secondly, we discussed how isotopic fractionation influences  $\delta^{34}S(OCS)$  values for atmospheric OCS as follows: all isotopic fractionation constants by OCS degradation are negative, indicating that the  $\delta^{34}S(OCS)$  values can be increased by OCS degradation in the troposphere. Because the main OCS sink is photosynthesis by plants, the  $\delta^{34}S(OCS)$  values in the atmosphere might be increased during the growing season in April. However, because of the long lifetime of OCS, the changes in  $\delta^{34}S(OCS)$  values might not be detected with a seasonal pattern. Future studies must be conducted for determination of isotopic fractionation constant and observation of  $\delta^{34}S(OCS)$  values to estimate the dynamics of atmospheric  $\delta^{34}S(OCS)$  values in troposphere.

Action: We added isotopic fractionation constants for OCS degradation process in our revised manuscript and discussed the matter in the revised manuscript ( $1^{st}$  and  $2^{nd}$  paragraph at section 3.5).

**References:**

Angert, A., Said-Ahmad, W., Davidson, C., and Amrani A.: Sulfur isotopes ratio of atmospheric carbonyl sulfide constrains its sources, Scientific Reports, 9(741), 1-8, 2018.

**Page 10, line 29 – These "further investigations" seem critical to establish the method. Are there new results since the manuscript was first submitted?**

**Reply:** Yes, we conducted further OCS preservation testing and added the results to section 3.4 in the revised manuscript. We added the results as follows: A rapid OCS decomposition of approximately 20 % during 7 days of storage was observed for the stainless steel adsorption tubes stored at 25 °C. A similar pronounced loss was observed for the Sulfinert-treated adsorption tubes stored at 4 °C but at a storage temperature of -20 °C. The OCS was stable for 30 days at -20 °C, and for at least 90 days at -80 °C within 1 $\sigma$  uncertainty of 6 % (Figure 8a). Furthermore, we found that the  $\delta^{34}$ S(OCS) values showed no significant change during storage for at least 14 days at -80 °C (Figure 8b). These results demonstrate that it is possible to apply this method for field campaigns by storing the adsorption tube at -80 °C after sampling.

Page 11, line 26 – Please be more specific. Instead of "some shortcomings in terms of sample amount" write down the ratio of sample amounts in the two methods.

**Reply:** We apologize for an ambiguous sentence.

Action: We added "IR-MS method requires a 300 times larger sample OCS than GC/MC-ICP-MS method." to section 3.6.

Page 11 - The conclusion section mentions for the first time another method (by Said-Ahmad, 2017). The mentioning of other existing methods should be done in the introduction, and comparison with other methods should be in the discussion, and include advantages and disadvantages. Similarly, writing in the last lines of the paper that the current method has the advantage of enabling carbon and oxygen isotopes measurements seems out of place, since this is not demonstrated in the current manuscript. However, this can be mentioned in the discussion.

**Reply:** We agree with your suggestion.**

Action: We added the GC/MC-ICP-MS method to the *Introduction* section. We made section 3.6 to present a comparison between IR-MS method and MC-ICP-MS method. Although our IR-MS method has shortcomings related to the sample size, we emphasize that this IRMS method can potentially be updated to multiple isotope measurement with carbon and oxygen isotopes. Also, we deleted the discussion related to carbon and oxygen isotope of OCS from the *Conclusion* section.

The discussion should also refer to the new paper by Angert et al., mentioned in the online discussion. Are the atmospheric values reported by the two papers identical, considering all experimental uncertainties? If not, is this a methods issue, or a real geographic effect?

**Reply:** As you have recommended, we cited Angert et al. (2018) in the revised manuscript. The  $\delta^{34}S(OCS)$  values of 13 ‰ reported by them are not the same as ours. Because our data are calibrated with the IAEA standard with to obtain  $\delta$  value with VCDT scale via chemical conversion from OCS to SF6, we do not expect that a 3 ‰ difference originates from some method-related difficulties in our system. To clarify whether or not we have method-related problems, inter-laboratory calibration is expected to be helpful for future studies.

Additionally, the geographic effect might induce differences of the  $\delta^{34}S(OCS)$  value between the two studies. In the revised manuscript, we discussed geographic effects that can be considered for explanation of the variations in  $\delta^{34}$ S(OCS) values. To discuss geographic effects, we added this discussion at 4th paragraph in section 3.5 and added a new citation as a relevant reference: Zumkehr et al. (2018).

**Reference**

Zumkehr, A., Hilton, T. W., Whelan, M., Smith, S., Kuai, L., Worden, J., Campbell, J. E.: Global gridded anthropogenic emissions inventory of carbonyl sulfide, Atmos. Environ., 183, 11-19, 2018.

Figure 3a – Maybe better to show versus sampling time (and not run number). Also the 10min is black in the legend and gray in the figure.

**Reply:** Thank you for your suggestion. We believe that the combination of Figure 3a and 3b nicely presents that there was no memory effect, and the relation between sampling time and OCS amount collected. Therefore, we retain this in the revised manuscript, but we changed the legend color from black to gray in the legend of Figure 3.

Action: We changed the legend color from black to gray in the legend of Figure 3.

Figure 8 – Better to start the y-axis at 80%. Also, need to show longer preservation periods.

Reply: Thank you for your suggestion. We did not start at 80 % on the y-axis because we added the OCS preservation results. We added results of longer preservation tests of OCS storage in adsorption tubes.

Thank you for reviewing our manuscript.

Shohei Hattori on behalf of co-authors.

---

## Author Comment (AC3)

Reply to RC2 from Referee #2

Blue: Referee comments
Black: Our comments
Red: The sentences in our manuscript

This manuscript attempts to address a current technical challenge within the atmo- spheric S and to some extent C cycle research community. It is clear that quantifying the movement of sulphur between the ocean, atmosphere and biosphere and detailing potential interactions and transformations is highly desirable but unfortunately it is still an extremely difficult task to conduct. Carbonyl sulphide is the most abundant sulphur containing gas in the atmosphere and currently there is much debate about its current sources and sinks. Within this framework the use of COS isotopic tracers might provide insights on this problem and help close the COS mass budget. As pointed out in the manuscript this not only holds interest for the communities working on the S cycle and its interactions with climate through chemical reactions in the troposphere and strato-sphere but also those working on the carbon cycle, as COS is a close analog of $CO_2$ and as such moves between the atmosphere and the biosphere alongside $CO_2$ during various steps of the photosynthetic pathway.

**Reply:** Thank you for reviewing our manuscript and for recognizing the importance of this study. We believe that the investigation of isotopic composition for carbonyl sulfide (OCS) can improve understanding of sulfur and carbon cycle.

Consequently, the motivation to study COS isotopes is growing and there is a need to overcome a number of technical challenges that limit the application of these tracers to help address the scientific questions introduced above. The first problem is that COS is a trace gas (atmospheric concentration around 500 parts per trillion) with an extremely low abundance of the rare isotope species thus large sample volumes are required for Isotope Ratio Mass Spectrometry (around 500 L providing ~10nmol of COS, in the case of the present study). Secondly, COS is an extremely reactive gas that can be hydrolyzed and/or produced rapidly from many surfaces and materials commonly used in gas exchange techniques.

**Reply:** As you have pointed out, collection of OCS from air and potential interface from preservation/loss of OCS during sampling are important for isotope analysis of OCS. For this study, we specifically examined how to collect OCS from air. We verified OCS contamination and OCS loss during measurement. This study is expected to be useful for situations in which large amounts of OCS, such as measurement of carbon and oxygen isotopic ratios of OCS, are required in the future, as we have described in our manuscript.

The current study presents a large volume sampling system adapted from a previous system designed for the measurement of volatile halocarbons and hydrocarbons by one of the co-authors of the present study (Bahlmann et al., 2011). This trapping system seems to be capable of tackling the first problem described above by trapping high volumes of air over a relatively short period of time (~100 minutes) and concentrating the air sample in a cryoshipper volume cooled previously with liquid $N_2$. Details regarding the temperature of the cryoshipper volume are a little vague stated as less than -110oC. This could be a bit more specific as pointed out by one of the reviewers as other gases may condense at lower temperatures.

**Reply:** We are sorry for the ambiguous sentence. Although it is difficult to keep the temperature constant using vapor of liquid $N_2$, we confirmed that the temperature in the dewar was –140 °C to –110 °C for at least 3 h. Because the sampling tube temperature was higher than –140 °C, $O_2$ is not trapped in the sampling tube during sampling. We increased the temperature during sampling.

**Action:** We added an explanation of the temperature range of the sampling tube to our revised manuscript.

The second problem of sample contamination during sampling and sample storage seems to still be an issue that would prevent the routine use of this sampling approach. For instance the authors purge the sampling tubes with high-purity Helium for 12H before sampling and require high temperatures starting at 160oC and reaching 330oC to condition the material. Can this be done in the lab beforehand on a batch of sampling tubes and does the surface remain inert thereafter, if so for how long? Is this step necessary when switching to Sulfinert valves and tubing? I am also curious to know how long roughly it takes to process one sample from the pre-conditioning stage to the final analysis of the purified sample on the IRMS?

**Reply:** We recognize that OCS contamination can be a serious issue. Indeed, although Tenax TA is suitable for OCS collection because of high-temperature conditioning, several picomoles of OCS are always observed from Tenax TA as blank. As you have pointed out, to reduce OCS blank from adsorbent as much as possible, we conditioned sampling and adsorption tubes for 6 h using pure helium (99.99995 % purity). The conditioning of sampling and adsorption tubes can be prepared in the laboratory before the field observation. We confirmed that the surface was inert for at least three days and the inactive state of the surface of adsorbents in these tubes would be retained under a no leakage condition. However, at present, we recommend conditioning immediately

before use to the greatest degree possible. Note that the conditioning steps would be required if stainless tubes are replaced by Sulfinert-treated tubes/valves, because the aim of conditioning is to include removal of strongly adsorbed volatile organic compounds such as ethanol and acetaldehyde in adsorbents. Given that these stored in the adsorption tube might block OCS collection and react with OCS (Ferm, 1957), we must remove these compounds as much as possible.

Finally, the time span is the following: sampling for 100 min (500 L), transfer for 40 min, pre-concentration for 40 min, and measurement using Q-MS or IR-MS for 20 min.

**Action:** We added information related to conditioning to section 3.3.

*It was useful to see the change in COS concentration during the preservation period. However, if I understand correctly this is not the experimental set-up for the rest of the data presented in the paper. Can you also plot the preservation data timeline for the non sulfinert experimental set-up please?*

**Reply:** As you have understood, we did not use Sulfiner-treated adsorption tubes for most of the experiments. As we added to Figure 8 in the revised manuscript, the OCS amount in a stainless steel adsorption tube stored at room temperature preserved $(-6 \pm 6)$ % of OCS, with no significant changes in $\delta^{34}S(OCS)$ values within $(0.2 \pm 0.4)$ ‰ after 3 h. All data sets are measurements taken right after the sampling (i.e. shorter than 30 min.). Therefore, we did not expect significant changes in OCS concentrations and the $\delta^{34}S(OCS)$ values.

**Action:** We added this information in the revised manuscript in section 3.4.

*Reviewing this as a method paper I feel there were a number of details missing or communicated a bit ambiguously. I am not sure it would be so easy to reproduce this methodology as a result. In particular section 2.1 was rather vague in several places, especially when it came to details of the calibration cylinders being used. Nowhere is the material of any of the calibration cylinders stated. This is not trivial as it is well known that COS is highly reactive and the use of stainless steel or aluminium cylinders for storage of COS standards will cause a drift in the COS concentrations over time and I would assume the isotopic composition too. For example, you state that the calibration cylinders F and G are much lower than that of atmosphere (e.g. G = 160 pmol mol-1), despite being sampled from the atmosphere that should be in the range of 350-500 pmol mol-1. I can only assume that contamination has occurred in the bottles during storage or a scrubber has been used whilst filling the tanks. If I understand these cylinders were filled in 2012 and assumed to represent the global background atmospheric composition. I am keen to know when the calibration curve described in 2.4.1 was actually completed a few days, months or years (2012) before the use of the calibration gases to validate the field measurements for COS concentration and δ34S? Overall calibration cylinders should be monitored closely over time when filled to see if the cylinder has issues and thereafter regularly checked for drift. It is also recommended to use Sulfinert cylinders or ACULIFE cylinders for COS. Can you confirm whether your calibration cylinders have special wall treatments to minimize contamination of your atmospheric COS gas?*

**Reply:** We are sorry that these ambiguous sentences have led to confusion. We found that there are three questions in this comment. First is for the material of cylinder. Second is for how to ensure OCS concentration if the OCS decomposed in cylinder. Third is for trends of OCS decomposition in the cylinder.

First, all cylinders are made of manganese steel without special wall treatments to minimize contamination.

Secondly, as you have expressed, the OCS concentrations and sulfur isotopic compositions for OCS can be changed in some cylinders. This is also a reason why we developed this method. To estimate OCS concentration, the OCS concentrations for sample A and sample B were calibrated using diluted in-house synthesized OCS (i.e. 100 %) to 10 % by high-purity He using a vacuum line. We confirmed that the OCS concentration and isotopic composition for sample B had not changed for four years after we published Hattori et al. (2015). Therefore, the calibration curved was made by using sample B and sample B was used as the daily working standard for sulfur isotopic measurement. The OCS samples in compressed air in the cylinder, on the other hand, were not stable: we found that the OCS concentration in sample H was decomposed to one third within three months. For that reason, we must conduct experiments immediately after we determined OCS concentrations. Therefore, for the example to make Figure 3, OCS in sample F were collected within two days to evaluate collection efficiency, followed by the determined OCS concentration in sample F within a week by calibration with sample B. In a similar manner, the cylinders of sample H, I, J, and K were used for experiment within 2–3 days. Therefore, we do not expect the changes in OCS concentration during the experimental period. Indeed, the effect OCS decomposition was not shown in Figure 3 or Figure 8a. We again emphasize, as you have stated, that the OCS contamination and its sulfur isotopic composition is not likely to be preserved in the cylinder. Also, no scrubber is used when the compressed air was filled in the cylinder.

Thirdly, the OCS concentration in sample B is monitored every time before the experiment. However, we do not monitor the OCS concentration of compressed air in cylinder because we know that the OCS do not preserve the $\delta^{34}S(OCS)$ value. As you suggest, we will plan to use a Sulfinert-treated cylinder and ACULIFE for the standard sample. Thank you for informing us.

To the revised manuscript, we added the following information in section 2.1 and 2.4:

・ Cylinder Material: "manganese steel without special wall treatments" .

・ Time span for measurement for sample F, H, I, J and K .

・ How we determined OCS concentration in sample A and B: " The OCS concentrations for samples A and B were determined against to the in-house synthesized OCS (i.e. 100 %) diluted to 10 % using high-purity He (99.99995 % purity; Japan Fine Products Co. Ltd.). It is noteworthy that the OCS concentration in sample B had showed no change at least four years after the publication of Hattori et al. (2015)."

I also think it would be worth discussing a little the caveats surrounding the fragmentation IRMS approach for example the potential for $O_2$ contamination and consequences for the $\delta^{34}S$ and $\Delta^{33}S$ analysis/interpretation.

**Reply:** Thank you for your suggestion. However, the caveats of fragmentation IRMS have already been discussed by Hattori et al. (2015). To avoid duplication, we did not add that information to the revised manuscript. As we described in that earlier report by Hattori et al. (2015), the influence by $O_2$ contamination to $\delta^{34}S$(OCS) value was also discussed; the $\delta^{34}S$ value are increased with the OCS amount depleted. The trend can be caused not by $O_2$ contamination because of natural abundance of rare oxygen isotope (see Hattori et al. (2015) for more detail). Although we must consider isotopic fractionation in ion source in IRMS and its sample size dependency, the effects of size dependence are negligible over 6 nmol of OCS in this study. We added points raised here.

Finally I think section 3.5 breaks the flow of the paper at the end and think it should rather become section 3.3. This would allow the paper to move out of the technical discussion into the scientific discussion and conclude more naturally.

**Reply:** We agree with your suggestion. We moved the results of preservation to section 3.4.

I also feel that the first paragraph of the conclusion is repetition of the results and should be removed.

**Reply**: Thank you for your suggestion. However, the summary of results is important for this method paper. For that reason, we changed the section title from "*Conclusion*" to "*Summary*".

I also agree with the other reviewer that the last paragraph introducing the carbon and oxygen isotopes should not suddenly appear here a bit out of the blue, their measurement is not trivial.

**Reply:** Agreed. We moved the discussion of carbon and oxygen to the *Discussion*.

Other details
The application of parentheses throughout this paper needs correcting, just because you quote a number does not mean it should be wrapped in brackets especially if it is integral to the sentence structure e.g. Page 1 line 26 and 27 highlights the type of problem that pops up throughout the manuscript. Please double check all parentheses are appropriate.

**Reply:** We checked all parentheses. We corrected parentheses as much as we can.

Page 1 line 26-27 How can you prove which value is correct? They both seem to be within the range of values for S sources. I would also try to clarify this argument/sentence a bit better.

**Reply:** As you have noticed, the both $\delta^{34}S$(OCS) value are in the range of expected $\delta^{34}S$(OCS) source. However, $\delta^{34}S$(OCS) value of 4.9 ‰ for compressed air in cylinder might be affected by decomposition in the cylinder and contamination from the compressor. However, $\delta^{34}S$(OCS) value of 10.5 ‰ is collected atmospheric OCS directly. To clarify this point, we added the "previous values of $\delta^{34}S$(OCS) = (4.9 ± 0.3) ‰ of compressed air in the cylinders were not representative samples for a global signal." to the revised text.

Page 2 Line 8 remove "net ecosystem exchange into" from this sentence.
**Reply:** Corrected accordingly.

Page 3 Line 11 state the purity of the Helium in %
**Reply:** We added the text "99.99995 % purity" to the revised manuscript.

Page 3 Line 12 can you provide more details of the exact S compounds used
**Reply:** We apologize for the ambiguous information. We used commercial sulfur powders. We added information related to sulfur powders and purity in our manuscript.

Page 3 Line 16 what type of commercial cylinders provide details and how often they were measured.
**Reply:** We used manganese steel cylinders without special wall treatment. The OCS concentration in sample A were measured every month. The OCS concentrations in sample B were measured before every experiment. Other OCS concentrations of compressed air in the cylinder were measured once except for the preservation test.

Can you also provide details of the compressor used to fill the bottles and any filters that were used on the compressor? Was the air dried before filling the cylinders?
**Reply:** We added information related to the compressor in section 2.1. The filter was not used for filling compressed air in a cylinder. The compressed air was not dried.

Page 3 Line 20 this is a super vague statement can you provide details of how you checked for stability.
**Reply:** We are sorry for including this unclear sentence. We deleted the sentence from the revised text because of duplication. Moreover, the method for preservation test was described in section 3.4.

Page 4 Line 3-4 can you explain how the heating was made and where was this step made in the lab or outside?
**Reply:** We conditioned using heating mantles in the laboratory.
**Action:** We added information to 3$^{rd}$ paragraph of section 2.2.

Page 4 Line 15 removed using what?
**Reply:** We removed the sampling tube from liquid nitrogen by hand.
**Action**: We added "manually" to this sentence (5$^{th}$ paragraph of sectin 2.2).

Page 4 Line 27-28 is the tubing flexible i.e. in a coil or rigid?
**Reply:** The tubes are made of stainless steel, but we can bend them.
**Action:** We added the U-shaped or coil-shaped before the trap.

Page 4 line 29 how is the heating achieved?
**Reply:** We used heating mantles for heating trap 2.
**Action:** We added information about the heater to 1$^{st}$ paragraph of section 2.3.

Page 4 line 30 why use steel and not glass. I guess this was later switched to a sulfinert tube?
**Reply:** There are three reasons for using stainless steel for traps. Glass also adsorbs OCS, stainless steel is easy to handle compared to glass. Also, the stainless steel has higher thermal conductivity than glass. We will change stainless steel tubes to Sulfinert-treated tubes in future work.

Page 5 line 8 this should be -72$^{\circ}$C
**Reply:** Corrected accordingly.

Page 5 Line 11 where did this injected gas come from?
**Reply:** We are sorry for the lack of information.
**Action:** We modified the sentence as follows: The retention times of $CO_2$ and OCS were initially determined by injecting a mixture of 8 mmol of $CO_2$ from pure $CO_2$ in a cylinder (99.995 % purity; Japan Fine Products Co. Ltd.) and 10 nmol of OCS from sample C in 2$^{nd}$ paragraph of section 2.3.

Page 5 lines 22-25 not written very clearly please improve
**Reply:** We apologize for this unclear sentence. We think that it is difficult to understand why we made two kinds of calibration curves and how we made calibration curves.
**Action:** We rewrote the information in 2$^{nd}$ and 3$^{rd}$ paragraph of section 2.4.

Page 5 line 25 how many replicates were analysed to obtain the precision?
**Reply:** We injected sample B in triplicate for each amount of sample size. The precisions were estimated by standard deviation of relative error between measured values and values estimated from calibration curves. The obtained precision was ± 3%.
**Action:** We added "$n = 3$" and "The precision (standard deviation ($1\sigma$) relative to mean) of the OCS amount by a syringe injection was estimated ± 3 % by the standard deviation of the relative error between the measured values and the estimated value for calibration curves." in 2$^{nd}$ paragraph of section 2.4.

Page 6 line 11 size of capillary tube?
**Reply:** We added capillary tube size information here.

**Reply:** We used a rotary pump (Pascal 2010; Pfeiffer Vacuum GmbH, Aβlar, Germany). Although we do not know the flow rate and pressure of pumping, we evacuated He gas at approximately 0.3 kPa per second from 30 cm$^3$ by regulating the seal valve (SS-4TW; Swagelok Co., Ohio, USA).
**Action:** We added the following to our manuscript: by a rotary pump (Pascal 2010; Pfeiffer Vacuum GmbH, Aßlar, Germany) gently with regulation using a valve.

**Reply:** The tank is sample B. We added the information to this sentence.

**Reply:** We apologize for the unclear text. We thought that it would be difficult to identify the OCS blank.
**Action:** We changed "OCS blank" to "OCS contamination".

**Reply:** We apologize for the unclear sentence.
**Action:** We rephrased the text to the following: In the developed system, the possibility exists that OCS is lost by passing OCS through GC1. Also, because the flow rate of approx. 50 mL / min was lower than the flow rate of approx. 200 mL / min reported by Hattori et al. (2015), the possibility exists that OCS was lost by Trap 1. Therefore, to assess these possibilities, the following test was conducted. Firstly, 5 nmol of OCS was injected to a system consisting of Trap 2, GC2, and Trap 4 and measured as true value. Then, the same amount of OCS was introduced into the developed purification system and the amount of OCS obtained was compared to true value.

**Reply**: We conducted the experiment manually. We changed slowly into "over 30 min by syringe" in this part.
**Action:** We added "manually" and "over 30 min by syringe" in our manuscript.

**Reply:** We used the OCS blank as OCS contamination. Therefore, we changed "OCS blank" into "OCS contamination"

**Reply:** We are sorry for mistakes in the use of units. We changed the "500 nmol mol$^{-1}$" to "500 pmol mol$^{-1}$".

**Reply:** The compressed air in the cylinder is collected just as air collected by the compressor. For that reason, the air was not dried. Additionally, Kamezaki et al. (2016) reported not hydrolysis but abiotic OCS decomposition.
**Action:** We deleted "OCS is decomposed by hydrolysis, which increases the $\delta^{34}$S(OCS) value." from our revised manuscript.

**Reply:** We apologize for the unclear sentence. We thought that we did not show the proportion to what.
**Action:** We changed the sentence in 1$^{st}$ paragraph of section 3.5 as follows: The $\delta^{34}$S(OCS) value of (10.5 ± 0.4) ‰ is generally consistent with earlier estimation by Newman et al. (1991), which expected the $\delta^{34}$S(OCS) values based on the flux of proportional to the organic matter were produced by photosynthesis as 3 ‰ and oceanic emission as 18 ‰ (Newman et al., 1991).

**Reply**: Thank you for the comment. For statistical analyses, we used $p$-tests. We assumed that the averaged OCS concentrations and $\delta^{34}$S(OCS) values in day and night are not significantly different. The calculated $p$-values are 0.65 and 0.29, for OCS concentrations and $\delta^{34}$S(OCS) value, respectively. These values are over 0.05. Therefore, significant differences are not observed for OCS concentrations and $\delta^{34}$S(OCS) values of day and night. To the revised manuscript, we added $p$-values.

**Reply:** We added all data accumulated from now to Figure 8 in section 3.4.

Thank you for reviewing our manuscript.

Shohei Hattori on behalf of the authors.

---

## Author Comment (AC4)

Reply to RC3 from Referee #1

Blue: Referee comment
Black: Our comment
Red: The sentences in our manuscript

To avoid unnecessary duplication, I will restrict my comments to aspects not mentioned by referees #2 and #3 yet.
**Reply:** Thank you for reviewing our work. We have revised the manuscript according to your comments.

1. The "large volume" aspect needs to be specified – clearly it refers to large volumes of air. Given that only the $^{34}S/^{32}S$ ratio could be analysed successfully, the title of the paper should be changed to "Large volume sampling system for measuring the $^{34}S/^{32}S$ isotope ratio of atmospheric carbonyl sulfide", or something along these lines.
   **Reply:** Thank you for comment. We changed "large volume" to "large volume air" throughout the revised manuscript; the title was changed to "Large volume of air sampling system for measuring the $^{34}S/^{32}S$ isotope ratio of atmospheric carbonyl sulfide" accordingly.

2. Section 2.4.1 should be renumbered 2.4 and renamed "Determination of the OCS concentration".
   **Reply:** This has been corrected accordingly.

3. Section 2.4.2 should be renumbered 2.5 and renamed "Determination of the sulfur isotopic composition of OCS".
   **Reply:** We have corrected this accordingly.

4. Table 2: You should include the results for the sulfur isotope deltas of samples A, F, G and H in the left hand column of this table, for ease of reference. Possibly, you could also present them in a separate table, given that sample G was analysed by Hattori et al. (2015) already, but gave a different result.
   **Reply:** To add $\delta^{34}S(OCS)$ values for each sample in these tables, we modified Table 1 and Table 2. We added the $\delta^{34}S(OCS)$ values of samples A–E in Table 1. The $\delta^{34}S(OCS)$ values of compressed air samples F, G, I, J, and K are presented in Table 2.

5. Table 2: Given that samples B, C and D all seem to have drifted with respect to the reference sample A, how did you ensure that the composition of sample A itself has not changed compared to the previous 2015 paper?
   **Reply:** We corrected $\delta^{34}S(OCS)$ values of the sample A to VCDT notion again to perform this study. Therefore, after regarding the comments, we compared the $\delta^{34}S(OCS)$ values between this study and that conducted by Hattori et al. (2015) for samples A and B. We added descriptions of how to correct the $\delta^{34}S(OCS)$ values to the VCDT notion.

   First, we determined the $\delta^{34}S$ value of sample A by converting OCS to $SF_6$; the $SF_6$ was measured for $\delta^{34}S$ relative to the VCDT scale by calibrating against $SF_6$ similarly converted from IAEA-S-1 ($Ag_2S$: $\delta^{34}S = -0.30$ ‰; Robinson, 1993) to $SF_6$, as described in Hattori et al. (2015). The $\delta^{34}S$ value of sample A was 12.6 ‰, which was 1.6 ‰ lower than the data presented in Hattori et al. (2015) with 14.2 ‰. The $\delta^{34}S$ value of sample B, that was used as a working standard for $\delta^{34}S$ measurements, was determined by comparison with the $\delta^{34}S$ value (in VCDT scale) of sample A. The $\delta^{34}S(OCS)$ value of sample B was $(14.1 \pm 0.2)$ ‰ in this study, showing no significant difference with the $\delta^{34}S(OCS)$ value of sample B $(14.3 \pm 0.2)$ ‰ reported by Hattori et al. (2015). Additionally, the OCS concentration in sample B remained unchanged. Therefore, sample B was used as a daily working standard to determine the $\delta^{34}S(OCS)$ values for all other samples (see Table 1 in the revised manuscript). For samples C and E, in-house OCS by reacting S powder with CO, were different from samples C and E presented by Hattori et al. (2015). However, it is noteworthy that samples C–E examined in this study were different batches of the experiment with Hattori et al. (2015) and not comparable. We regret the confusion this has caused.

   We agree on the need to clarify how we determined $\delta^{34}S$ relative to VCDT for samples. According to the discussion raised above, we added related explanations to section 2.5 in the revised manuscript.
   **Reference**

Robinson, B. W., Sulfur isotope standards, Reference and inter comparison materials for stable isotopes of light elements, in Proceedings of a consultants Meeting Held in Vienna, 1–3 December, 39–46, 1993.

6. 5/31: One could hypothesis that samples F, G and H all started out at the same initial OCS mole fraction and isotope ratio. Adopting this hypothesis, could you please include a plot of their isotope deltas vs. the natural logarithm of the "residual" OCS fraction (i.e. a Rayleigh fractionation plot) to check whether the apparent OCS loss in the cylinders follows a common fractionation constant $\varepsilon$?

   **Reply:** We tried your suggested calculations. However, we did not measure the OCS concentration in samples H, I, J, and K using glass bottles directly. We measured OCS concentration after sampling. Although the OCS concentration measured after sampling might not be robust, we measured the OCS concentration roughly. When we assumed the original OCS concentrations for samples F, G, and K to be the same as sample J, which had the highest OCS concentration, the relation is not on the Rayleigh plot, indicating the samples do not follow a common fractionation constant and/or the origin of OCS concentration and $\delta^{34}S(OCS)$ values are not the same.

[Figure]

Figure R1: Relative $\delta^{34}S(OCS)$ value relative to the natural logarithm of the residual OCS fraction. We assumed sample J have original OCS concentration and $\delta^{34}S(OCS)$ value.

7. 7/26: Please describe in detail how you introduced these aliquots of sample B?

   **Reply:** Sample B was injected manually from the syringe injection port, which is tee with septum equipped before the condenser, over 30 min.

   **Action:** We modified Figure 1 and described in detail in 1st paragraph of section 3.2 as the following: We introduced aliquots of 3 nmol, 6 nmol, 10 nmol, and 15 nmol of sample B over 30 min with a gastight syringe via a syringe port made from a tee union with a septum. The syringe port was place between the inlet filter and the condenser and the sampling inlet was connected to of high-purity $N_2$ ($>$ 99.99995 vol. %; Nissan Tanaka Corp., Saitama, Japan) (Figure 1).

8. 8/27: The $m/z$ 33 interference could also be due to $NF^+$ (e.g. from $NF_3$).

   **Reply:** Thank you for this comment. We agree that $NF_3$ is also a possible candidate, as you have suggested.

   **Action:** We added $NF_3$ to this sentence of the revised manuscript.

9. 11/25: Please state the precision achieved for OCS analysis in this earlier paper.

   **Reply:** We added a new section (section 3.6) to explain a comparison between our methods using GC/IRMS and GC/MC-ICP-MS. We added the comparison of precisions in that paragraph.

10. 22/9: The precision achieved for sample B is not meaningful for these air samples. Please replace the error bars with a more suitable estimate of the precision for an actual air sample.

    **Reply:** Thank you for your critical comment. We estimated the blank effect when the 5 % of contaminated OCS ranging $\delta^{34}S(OCS)$ value from 3 to 18 ‰ as follows. 5 % of OCS contamination change the accuracy of $\delta^{34}S(OCS)$ value with −0.3 to +0.3 ‰. The precision of our repeated measurement is ± 0.2 ‰. The overall precision of measurement is ± 0.4 ‰. Additionally, the standard deviation of four atmospheric samples we observed was ± 0.2 ‰. Therefore, the $\delta^{34}S(OCS)$ value for

atmospheric OCS at Suzukakedai campus is (10.5 ± 0.4) ‰. We modified the precision and Figure 6 in our revised manuscript.

11. Referee #2 commented on the use of parentheses in your manuscript. The notation "($x \pm s_x$) ‰" (and similar) follows in fact international guidelines on the SI such as NIST Special Publication 811 2008 Edition "Guide for the Use of the International System of Units (SI)" and the IUPAC Green Book, 3rd edition, p. 151 (section 8.1, example 2; http://www.iupac.org/fileadmin/user_upload/publications/e-resources/ONLINE-IUPAC-GB3-2ndPrinting-Online-Sep2012.pdf. As the journal advocates the use of the SI, no change is necessary.
**Reply:** Thank you for supporting our presentation of our work. Yes, for this point, we made revisions during the process of revision for AMTD.

**Technical corrections**

- 2/4: Brühl et al.
  **Reply:** Corrected.

- 2/11: S equivalents
  **Reply:** Corrected.

- 2/17: O($^3$P) – spin states are not written in italics
  **Reply:** Corrected.

- 3/24: compartments
  **Reply:** We changed "comportments" into "compartments".

- 3/25 & various occurrences elsewhere: Sulfinert
  **Reply:** We changed all cases of "Sulfinert®" or "sulfinert" into "Sulfinert" throughout the manuscript.

- 5/30: Samples F, G and H
  **Reply:** Corrected.

- 7/23: Add full-stop after system and start new sentence "We sequentially..."
  **Reply:** As commented also by other reviewers, the sentence was not clear.
  **Action:** We rewrote this sentence as 1st paragraph of section 3.2 in the manuscript as follows: In the developed system, the possibility exists that OCS is lost by passing OCS through GC1. Also, because the flow rate of approx. 50 mL / min was lower than the flow rate of approx. 200 mL / min reported by Hattori et al. (2015), the possibility exists that OCS was lost by Trap 1. Therefore, to assess these possibilities, the following test was conducted. Firstly, 5 nmol of OCS was injected to a system consisting of Trap 2, GC2, and Trap 4 and measured as true value. Then, the same amount of OCS was introduced into the developed purification system and the amount of OCS obtained was compared to true value.

- 7/25: dependence
  **Reply:** We changed "dependency" to "dependence".

- 7/31 & 32: Replace full stop after $\sigma$ with "uncertainty: ".
  **Reply:** This point was corrected accordingly.

- 8/26 to 8/28: Remove colon (:) after $m/z$ (e.g. $m/z$ 32)
  **Reply:** Removed all colons (:) with m/z.

- 9/5: USA
  **Reply:** Corrected accordingly.

- 11/5: proofed -> showed
  **Reply:** Corrected accordingly.

Thank you for reviewing our manuscript.

Shohei Hattori on behalf of co-authors.

---

## Author Response (AR2)

**Author responses**

Comment from Referee 1

——

5   The manuscript is now significantly improved, with revisions made in accordance with the reviewers suggestions. The only concern I have now is that the abstract was not updated according to the changes in the main text. This main text now states that the measured value of 10.5 may partly result from "Potential anthropogenic OCS sources are Chinese emissions from rayon (yarn and staple) and coal (industry and residential)", and that these emissions may explain the difference from slightly higher isotopic value published by Angert et al (by the way,

10   trying to look up this reference it turned out to be from 2019, and not 2018 as cited). However the Abstract still says "we instead propose of a d34S(OCS) value of (10.5 ± 0.3) ‰ for the tropospheric OCS". Please correct so the abstract will reflect the discussion. Also, there is no need to mention in the abstract twice, the value of 4.9 which is now considered an artifact.

——

15   Reply: We have corrected the abstract according to the comment.

Comment from Referee 2

—

Table 1 in my versions of the manuscript was incomplete (missing text and columns on the right). Unfortunately,

20   it was missing in both the "manuscript" and in the "author's response" version.

Angert et al. paper - The publication year is 2019.

—

Reply: Corrected accordingly.

Many thanks, Shohei Hattori on behalf of co-authors.

(marked-up manuscript)

[revised manuscript text omitted]